# Kinesin-like motor protein KIF23 maintains neural stem and progenitor cell pools in the developing cortex

Sharmin Naher[1,2], Kenji Iemura [ID][3], Satoshi Miyashita[4], Mikio Hoshino [ID][4], Kozo Tanaka [ID][3], Shinsuke Niwa [ID][5,6], Jin-Wu Tsai [ID][7,8,9], Takako Kikkawa [ID][2✉] & Noriko Osumi [ID][1,2✉]

## Abstract

**Accurate mitotic division of neural stem and progenitor cells (NSPCs) is crucial for the coordinated generation of progenitors and mature neurons, which determines cortical size and structure. While mutations in the kinesin-like motor protein *KIF23* gene have been recently linked to microcephaly in humans, the underlying mechanisms remain elusive. Here, we explore the pivotal role of KIF23 in embryonic cortical development. We characterize the dynamic expression of KIF23 in the cortical NSPCs of mice, ferrets, and humans during embryonic neurogenesis. Knockdown of *Kif23* in mice results in precocious neurogenesis and neuronal apoptosis, attributed to an accelerated cell cycle exit, likely resulting from disrupted mitotic spindle orientation and impaired cytokinesis. Additionally, KIF23 depletion perturbs the apical surface structure of NSPCs by affecting the localization of apical junction proteins. We further demonstrate that the phenotypes induced by *Kif23* knockdown are rescued by introducing wild-type human *KIF23*, but not by a microcephaly-associated variant. Our findings unveil a previously unexplored role of KIF23 in neural stem and progenitor cell maintenance via regulating spindle orientation and apical structure in addition to cytokinesis, shedding light on microcephaly pathogenesis.**

**Keywords** Kinesin Motor Proteins; Neurogenesis; Cytokinesis; Apoptosis; Microcephaly
**Subject Categories** Cell Cycle; Neuroscience; Stem Cells & Regenerative Medicine

## Introduction

The mammalian cerebral cortex is generated through meticulously orchestrated developmental processes, where neural stem/progenitor cells (NSPCs) play central roles in their proliferation, neuronal differentiation, and neuronal migration (Götz and Huttner, 2005). In the initial stages of corticogenesis, NSPCs within the ventricular zone (VZ) of the cortex undergo multiple rounds of proliferative division, thereby expanding the progenitor pool. As corticogenesis progresses, neurogenic division becomes predominant, giving rise to neurons either directly or indirectly via transit-amplifying progenitors, i.e., intermediate progenitors (IPs) (Kriegstein et al, 2006; Pontious et al, 2008). Disruptions in these finely tuned processes can result in abnormal brain size and structural anomalies.

The Kinesin superfamily motor proteins (Kifs) are responsible for controlling microtubule dynamics and facilitating intracellular transportation of organelles, mRNAs, and protein complexes along microtubules (Hirokawa et al, 2009). In the nervous system, members of Kif family are localized in both mitotic and post-mitotic cells. In post-mitotic neurons, Kifs are essential motor proteins involved in neuronal migration, transport, and synaptic transmission (Joseph et al, 2021). Dysregulated expression of KIFs has been implicated in several neurological disorders, including amyotrophic lateral sclerosis, epilepsy, and Alzheimer's disease (Asselin et al, 2020; Guillaud et al, 2020; Lucero et al, 2022). While the role of Kifs in neurons is well-established, their functions in NSPCs during embryonic brain development remain less explored.

Within the Kif family, the kinesin-6 subfamily, comprised of three members - Kif20a, Kif20b, and Kif23, are notably recognized for their pivotal role in mitotic processes (Hirokawa et al, 2009; Janisch et al, 2018). Specifically, Kif20a regulates the division of NSPCs by interacting with the cell cycle regulators in the intercellular bridge of dividing NSPCs (Geng et al, 2018; Qiu et al, 2020). Kif20b is required for midbody organization in dividing NSPCs during corticogenesis (Janisch et al, 2013). Meanwhile, kinesin-like motor protein Kif23, a prominent member of the Kif 6 family, assumes a critical role as a major component of the central spindle complex. During mitosis in cell lines, Kif23 conspicuously localizes at the cleavage furrow and the midbody, where it governs the bundling of anti-parallel microtubules and facilitates cytokinesis (Adams et al, 1998; Pavicic-Kaltenbrunner et al, 2007). In cultured neuronal cells, Kif23 serves as an essential

[1]Department of Developmental Neuroscience, Graduate School of Life Sciences, Tohoku University, 2-1-1 Katahira, Aoba-Ku, Sendai, Miyagi 980-8577, Japan. [2]Department of Developmental Neuroscience, Tohoku University Graduate School of Medicine, 2-1, Seiryo-Machi, Aoba-ku, Sendai, Miyagi 980-8575, Japan. [3]Department of Molecular Oncology, Institute of Development, Aging and Cancer (IDAC), Tohoku University, Sendai, Miyagi 980-8575, Japan. [4]Department of Biochemistry and Cellular Biology, National Institute of Neuroscience, NCNP, Tokyo 187-8502, Japan. [5]Graduate School of Life Sciences, Tohoku University, Sendai, Miyagi 980-8578, Japan. [6]Frontier Research Institute for Interdisciplinary Sciences (FRIS), Tohoku University, Sendai, Miyagi 980-0845, Japan. [7]Institute of Brain Science, College of Medicine, National Yang Ming Chiao Tung University, Taipei, Taiwan. [8]Department of Biological Science and Technology, College of Biological Science and Technology, National Yang Ming Chiao Tung University, Hsinchu, Taiwan. [9]Brain Research Center, National Yang Ming Chiao Tung University, Taipei, Taiwan. ✉E-mail: takako.kikkawa.c4@tohoku.ac.jp; noriko.osumi.c7@tohoku.ac.jp

factor in maintaining dendritic morphology and composition (Sharp et al, 1997; Yu et al, 2000). Notably, recent genetic studies in humans have identified mutations in human *KIF23* gene associated with microcephaly (Karaca et al, 2015; Boonsawat et al, 2019). However, the precise role of Kif23 in the intricate development of the mammalian cortex remains uncharacterized.

In this study, we uncovered Kif23's exclusive expression within the NSPCs, particularly in the spindle during early mitosis, alongside its established expression in the midzone and midbody. To delineate Kif23's in vivo role in NSPCs, we utilized the technique of in-utero electroporation in developing mouse embryos. Knockdown (KD) of *Kif23* disrupted the recruitment of cytokinesis initiation factors in dividing NSPCs, resulting in binucleation, followed by a cascade of events, including precocious neurogenesis, neuronal apoptosis, and a substantial reduction in neuronal production. Our study further revealed the distinct function of Kif23 in regulating spindle orientation and apical structure, setting it apart from other members of the kinesin-6 family. Notably, the introduction of human *KIF23* bearing a mutation associated with microcephaly failed to rescue the defects induced by *Kif23*-KD in the developing cortex. Thus, our findings offer novel insights into how KIF23 dysfunction in humans may contribute to microcephaly by uncovering previously unexplored cellular and molecular mechanisms of Kif23 in NSPC maintenance.

# Results

## Kif23 is highly expressed in the embryonic mouse neocortex

In order to elucidate the role of Kif23 in neocortical development, our initial investigation focused on the expression of *Kif23* within the developing neocortex. We leveraged the spatial transcriptomic dataset derived from the E15.5 mouse brain (Tsai et al, 2024; Data Ref: Tsai et al, 2024), which categorized gene expression profiles into 12 distinct clusters (Fig. 1A, left). Interestingly, Kif23 exhibited specific expression within cluster 10, a cluster primarily situated in the ventricular zone (VZ), where NSPCs are enriched (Fig. 1A, middle and right).

We next examined previously published single-cell RNA sequencing (scRNA-seq) data from E14.5 mouse cortex (Loo et al, 2019; Data Ref: Loo et al, 2019). Through a meticulous reanalysis of this scRNA-seq dataset, we stratified cell clusters into distinct categories: NSPCs (NS), IPs (IP), neurons (N), and interneurons (IN) (Fig. 1B, left). Remarkably, *Kif23* is shown to be enriched in the actively proliferating NSPCs, particularly prominent in the NS2 cluster expressing *Top2a*, a marker of proliferative cells (Fig. 1B, middle and right).

We indeed confirmed the abundance of *Kif23* mRNA in the VZ by in situ hybridization at E14.5 (Fig. 1C). Subsequently, we examined Kif23 protein expression in NSPCs by co-immunostaining with Sox2, a marker for NSPCs, and Tbr2, a marker for IPs (Fig. 1D; Appendix Fig. S1A). We found that a high proportion of Kif23+ cells co-localized with Sox2+ NSPCs (~90%), while Kif23 expression was also detected in a subset of Tbr2+ IPs (~20%) (Appendix Fig. S1B). During cortical development, Kif23 protein was detected at a higher level in the neocortical VZ progenitor cells at E12.5 and E14.5 during neurogenic stages, while its expression declined at E16.5 when the neurogenesis decreased (Fig. 1E). This stage-specific expression in NSPCs suggests a potential role of Kif23 in regulating early cortical neurogenesis.

## Knockdown of *Kif23* induces premature neuronal differentiation

Next, we employed an in-utero electroporation technique to knockdown (KD) *Kif23* by delivering a small interfering RNA (siRNA) targeting the mouse *Kif23* gene, together with an EGFP expression vector, into the mouse neocortex at E14.5, the same stage as the scRNA-seq dataset (Fig. 2A). Successful *Kif23* siRNA-mediated reduction of endogenous Kif23 protein expression was confirmed at E15.5 (Fig. 2B).

In this context, we examined the distribution of GFP+ cells across various zones of the developing cortex two days after *Kif23*-KD (at E16.5). Notably, the percentage of GFP+ cells in the VZ/subventricular zone (SVZ) exhibited a significant decrease in *Kif23*-KD cortices compared to the control group, whereas the proportion of GFP+ cells in the intermediate zone (IZ) significantly increased in the *Kif23*-KD cortices (Fig. 2C,E).

In light of the migration of NSPCs in the VZ/SVZ to the IZ and CP upon exiting the cell cycle, the observed perturbation in cell distribution hints at the potential induction of premature differentiation attributable to *Kif23*-KD. We thus conducted an analysis of the neuronal marker Tuj1 (βIII-Tubulin) expression within both the control and *Kif23*-KD groups. Remarkably, a significantly larger portion of GFP+ cells in the *Kif23*-KD cortices co-expressed Tuj1 in comparison to the control cortices at E16.5 (Fig. 2D,F). Furthermore, this trend was consistent with the elevated Tuj1 expression within the VZ of *Kif23*-KD cortices at E15.5 (Fig. 2G). Collectively, these findings substantiate the notion that *Kif23*-KD fosters premature neuronal differentiation in the developing cortex.

## Kif23 deficiency induces apoptotic cell death

Interestingly, our investigation unveiled a significant increase in number of GFP+ cells within the CP and IZ, along with a corresponding decrease in the VZ/SVZ three days after *Kif23*-KD (at E17.5) (Fig. EV1A,B). Moreover, a significant reduction in the overall GFP+ cell population was evident in the KD group (Fig. EV1A,C). This profound reduction of GFP+ cells suggests the involvement of additional underlying mechanisms beyond mere precocious neuronal differentiation. Subsequently, we conducted an examination of apoptosis in the E15.5 *Kif23*-KD cortices one day after electroporation. The results were striking, revealing a noteworthy increase in pyknotic nuclei, indicative of chromatin and nucleus condensation during apoptotic cell death, within the VZ, SVZ, and IZ, as detected by DAPI staining (Fig. 3A,B). Immunostaining with an apoptosis marker, cleaved caspase 3 (CC3), further confirmed our observation, highlighting the heightened apoptotic activity within *Kif23*-KD cortices in comparison to the control group (Fig. 3A,C).

To identify the specific cell types undergoing apoptosis, we performed immunostaining with antibodies against various markers. Notably, the pyknotic nuclei were predominantly observed in cells expressing neuronal markers Hu (~70%) and Tuj1 (~60%). In contrast, only a small proportion of pyknotic nuclei were detected

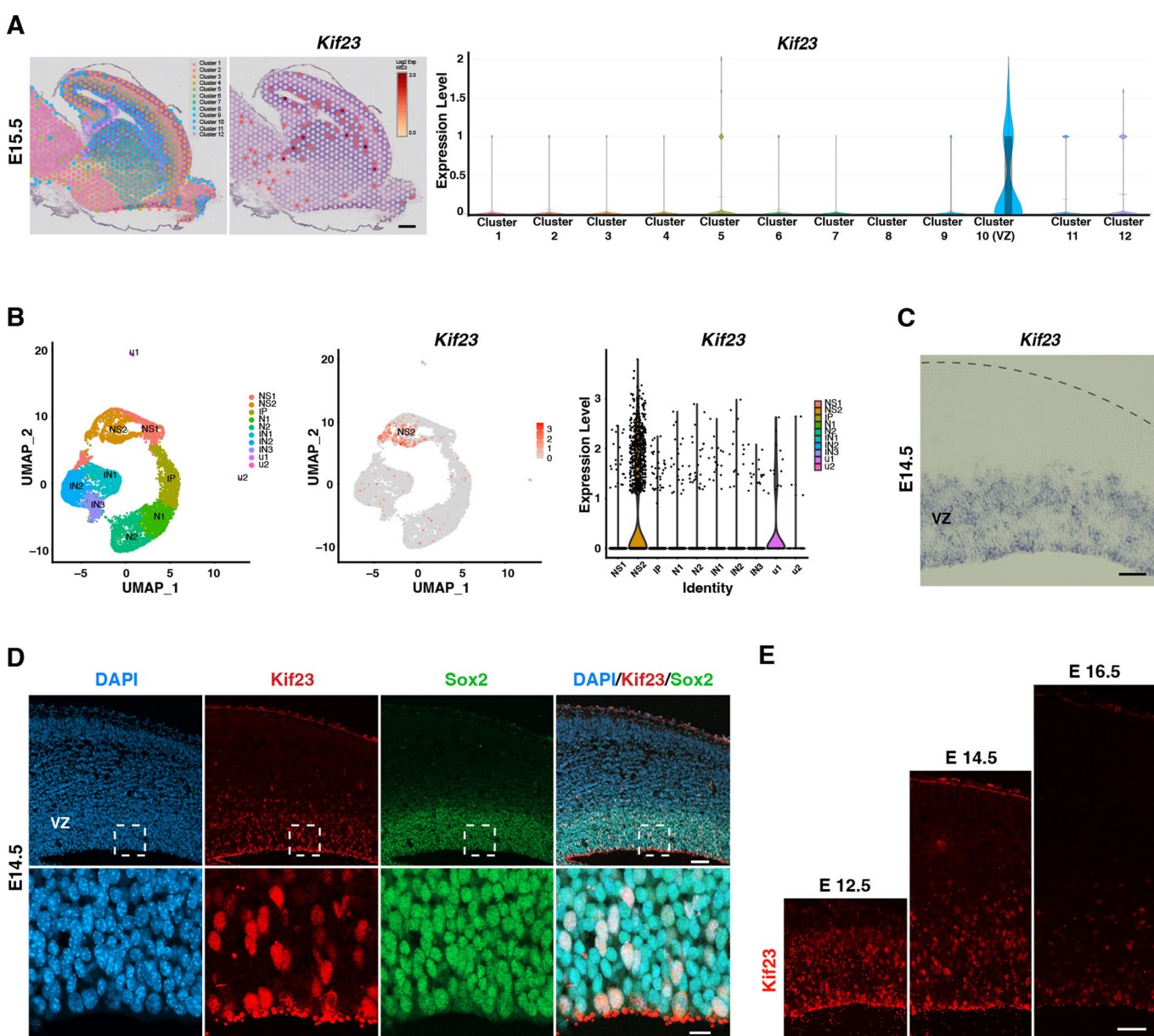

**Figure 1. Kif23 expression in the developing neocortex.**

(A) Visium spatial gene expression analysis of E15.5 mouse brain showing gene expression clusters (left), spatial plots (middle), and violin plots (right) to show the spatial expression of *Kif23*. The dataset was obtained from Tsai et al, 2024. *Kif23* is enriched in the cluster 10. Scale bar, 300 μm. (B) Single-cell RNA-seq analysis of E14.5 mouse cortex showing cell type clusters (left), feature plots (middle), and violin plots (right) to show the expression of *Kif23*. The dataset was obtained from Loo et al, 2019. *Kif23* is enriched in the neural stem/progenitor cells cluster 2 (NS2). NS1, neural stem/progenitor cells cluster 1; IP, intermediate progenitor cells cluster; N1 and N2, neurons cluster 1 and 2; IN1, IN2 and IN3, interneurons cluster 1, 2 and 3. (C) Expression of *Kif23* mRNA detected by in situ hybridization of E14.5 embryonic mouse cortex. *Kif23* is localized in the ventricular zone (VZ). Scale bar, 30 μm. (D) Representative image of E14.5 mouse cortex stained with the antibodies to Kif23 and Sox2, a marker of neural stem progenitor cells (NSPCs) and DAPI. Kif23 is co-localized with Sox2. Boxed areas are magnified in the bottom panel. Scale bars, 50 μm (top) and 10 μm (bottom). (E) Immunostaining of the mouse cortex at E12.5, E14.5 and E16.5 showing expression of Kif23 at a higher level in E12.5 and E14.5. Scale bar, 50 μm. Source data are available online for this figure.

in cells expressing an NSPC marker Sox2 (~25%) and an IP marker Tbr2 (~24%) within *Kif23*-KD cortices (Fig. 3D,E). These findings suggest that differentiating neurons are particularly susceptible to excessive apoptotic cell death following *Kif23*-KD. Further characterization of the apoptotic cells revealed the presence of pyknotic single cells with micronuclei and pyknotic doublets (~7%) (Fig. 3F,G). These micronuclei and pyknotic doublets closely

resemble binucleated cells, a phenomenon known to result from improper cytokinetic events (Tedeschi et al, 2020; Xie et al, 2021). Co-staining with Tuj1 and CC3 revealed that the majority of pyknotic doublets in the VZ were double positive for Tuj1 and CC3 (~76%), consistently indicating that these cells, which failed in cytokinesis, might die and be eliminated soon after differentiating into neurons (Appendix Fig. S2A,B). Co-staining with CC3 and a

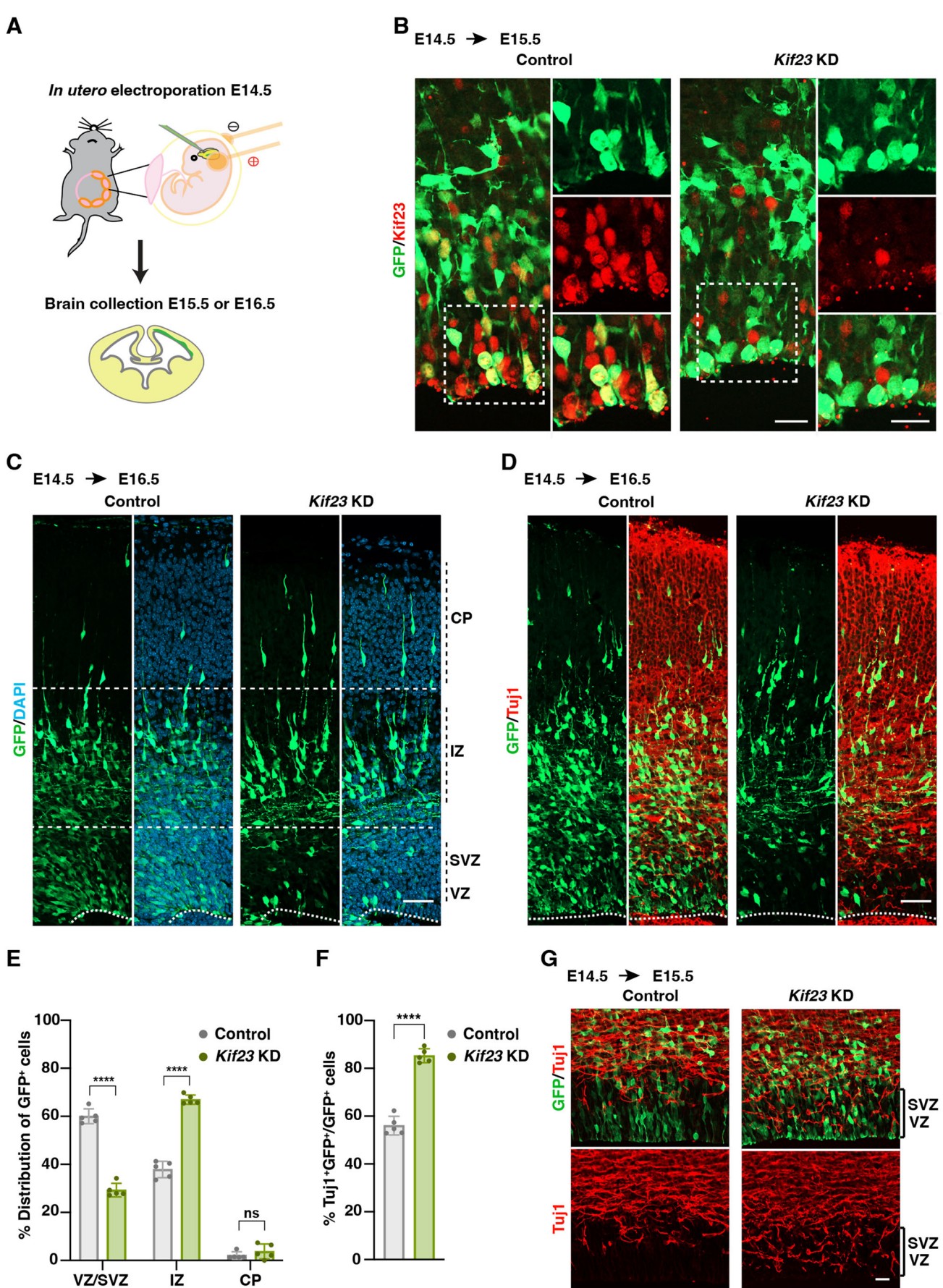

**Figure 2.   *Kif23* knockdown induces premature neuronal differentiation.**

(A) Schematic diagram of *Kif23* knockdown (KD) by in-utero electroporation. The embryonic mouse brains were electroporated at E14.5 with control or *Kif23* siRNA. (B) Representative images stained for GFP and Kif23 in the E15.5 mouse cortices after in-utero electroporation. The expression of Kif23 is effectively suppressed by *Kif23* siRNA. Boxed areas are magnified. Scale bars, 20 µm. (C) Representative images of the distribution of GFP$^+$ cells in the control and *Kif23*-KD cortices. Dashed lines illustrate the ventricular surface and the borders among the ventricular zone/ subventricular zone (VZ/SVZ), intermediate zone (IZ), and cortical plate (CP). Scale bar, 50 µm. (D) Representative images of the control and *Kif23*-KD cortices at E16.5 stained for GFP and Tuj1. Dashed lines illustrate the ventricular surface. Scale bar, 50 µm. (E) Quantification of the distribution of GFP$^+$ cells in VZ/SVZ, IZ, and CP, respectively within the 150 µm wide column of E16.5 cortices. The data represent the mean ± SD (Control $n = 874$ cells, 5 embryos; *Kif23*-KD $n = 433$ cells, 5 embryos). Two-way ANOVA with Bonferroni's multiple comparisons test, $p$ values from left to right: ****$p = 9.0E-15$, ****$p = 2.9E-14$, ns not significant. (F) Quantification of the percentage of Tuj1$^+$GFP$^+$ cells relative to the total GFP$^+$ cells within the 150 µm wide column of E16.5 cortices. The data represent the mean ± SD (Control $n = 894$ cells, 5 embryos; *Kif23*-KD $n = 408$ cells, 5 embryos). Two-tailed Student's $t$ test, ****$p = 9.6E-07$. (G) Representative images of the control and *Kif23*-KD cortices at E15.5 stained for GFP and Tuj1. Scale bar, 20 µm. Source data are available online for this figure.

neuronal marker NeuroD2 also showed that pyknotic cells were rarely positive for NeuroD2 (12.3% ± 3.4%, $n = 200$ cells, 4 embryos) (Appendix Fig. S2C). Considering the transient expression of NeuroD2 in relatively late differentiating neurons compared to Tuj1 (Sagner et al, 2021; Farah et al, 2000), it is possible that differentiated neurons undergo apoptosis before expressing NeuroD2. Consequently, our results put forth the intriguing notion that a subset of cells fails to complete cytokinesis in Kif23-deficient NSPCs, shedding light on the intricacies of Kif23's role in cell division and survival.

## Knockdown of *Kif23* inhibits proliferation of progenitor cells

On the basis of the increase in differentiated cells, we considered the possibility that Kif23 plays a vital role in the maintenance and proliferation of NSPCs. Notably, our findings revealed that *Kif23*-KD led to a significant decrease in the percentage of cells that co-expressed GFP and Pax6 (a NSPC marker), as compared to the control group (Fig. 4A,B). As NSPCs produce IPs, we assessed the impact of NSPC depletion on IP population. Similarly, the proportion of cells positive for both GFP and Tbr2 (an IP marker) was significantly lower in the *Kif23*-KD group in comparison to the control group (Fig. 4C,D). These observations clearly indicate that *Kif23*-KD depleted both apical and basal progenitor pools.

To gauge the impact of *Kif23*-KD on the overall proliferative capacity of NSPCs, we adopted a dual approach. Firstly, we electroporated the embryonic mouse cortex at E14.5 and subsequently administrated an intraperitoneal injection of EdU (labeling S-phase proliferating cells) just two hours before the sacrifice at E15.5 or E16.5. Remarkably, the percentage of EdU$^+$GFP$^+$ cells among the GFP$^+$ cells in the VZ, which serves as a reliable proliferative index, significantly reduced in response to *Kif23*-KD (Fig. 4E,F; Appendix Fig. S3A,B). We also assessed the mitotic activity by staining the electroporated cortices at E15.5 and E16.5 for a mitotic marker phospho-histone-3 (PH3). Notably, there was a significant decrease in the number of PH3$^+$GFP$^+$ cells in the VZ of *Kif23*-KD cortices compared to the control group (Fig. 4G,H; Appendix Fig. S3C–E). These compelling results collectively underscore the indispensable role of Kif23 in promoting the proliferation of NSPCs during the critical period of embryonic neurogenesis.

It is established that when the proliferation of NSPCs is inhibited, these cells tend to exit the cell cycle and differentiate into neurons (Qiao et al, 2018). We thus examined whether these *Kif23*-KD progenitors prematurely departed from the cell cycle. To

facilitate this analysis, we performed IUE of the embryonic mouse cortex at E14.5, followed by an EdU injection at E15.5. Subsequently, we examined GFP$^+$EdU$^+$Ki67$^-$ cells, at E16.5, which represent the cell cycle-exiting proportion. Our observation uncovered a significant increase in the percentage of GFP$^+$EdU$^+$Ki67$^-$ cells among the GFP$^+$EdU$^+$ cell population in response to *Kif23*-KD (Fig. 4I,J). This finding provides clear evidence that the loss of Kif23 promoted an early exit from the cell cycle. Therefore, our results consistently affirm the critical role of Kif23 in the maintenance and renewal of NSPCs.

## Kif23 regulates mitotic spindle orientation

Given the essential role of Kif23 in central spindle assembly and cytokinesis within HeLa cells (Mishima et al, 2004), we examined the subcellular localization of Kif23 protein across various phases of the cell cycle by assessing nuclear morphology via DAPI staining and the positions of a centrosomal protein γ-tubulin. Our observation revealed distinct Kif23 localization within the nucleus of NSPCs during G2/S, and prophase (Fig. 5A). However, as the cell cycle progressed, Kif23 was relocated to the spindle during prometaphase and metaphase, occupied the midzone during anaphase and telophase, and ultimately gathered within the midbody during the cytokinetic phase (Fig. 5A).

The dynamic shifts in Kif23 localization within the NSPCs, particularly its co-localization with spindle microtubule marker α-tubulin during mitosis (Fig. 5B), point toward Kif23's pivotal role in orchestrating the mitotic spindle. To further explore this issue, we examined morphology of mitotic spindles in cells containing visible spindle poles labelled with γ-tubulin and measured the fluorescence intensity of α-tubulin in the spindle areas. Our analysis revealed no noticeable difference in spindle organization and α-tubulin density following *Kif23*-KD (Figs. 5C,D and EV2), indicating Kif23 is not essential for formation and organization of mitotic spindles.

Given the function of spindle cleavage plane orientation as an essential determinant of NSPC fate (Yingling et al, 2008), our investigation delved into the orientation of the mitotic spindle, scrutinizing the cleavage plane's alignment in relation to the apical surface (Fig. 5E). Notably, *Kif23*-KD NSPCs displayed a larger spindle angle when compared to the control group (Fig. 5F). Additionally, the percentage of mitotic progenitors with spindle cleavage planes oriented horizontally (control, 10.94%; *Kif23*-KD, 27.37%) or obliquely (control, 21.87%; *Kif23*-KD, 36.84%) in relation to the apical surface increased after *Kif23*-KD, while the percentage of the those with vertically oriented spindle cleavage

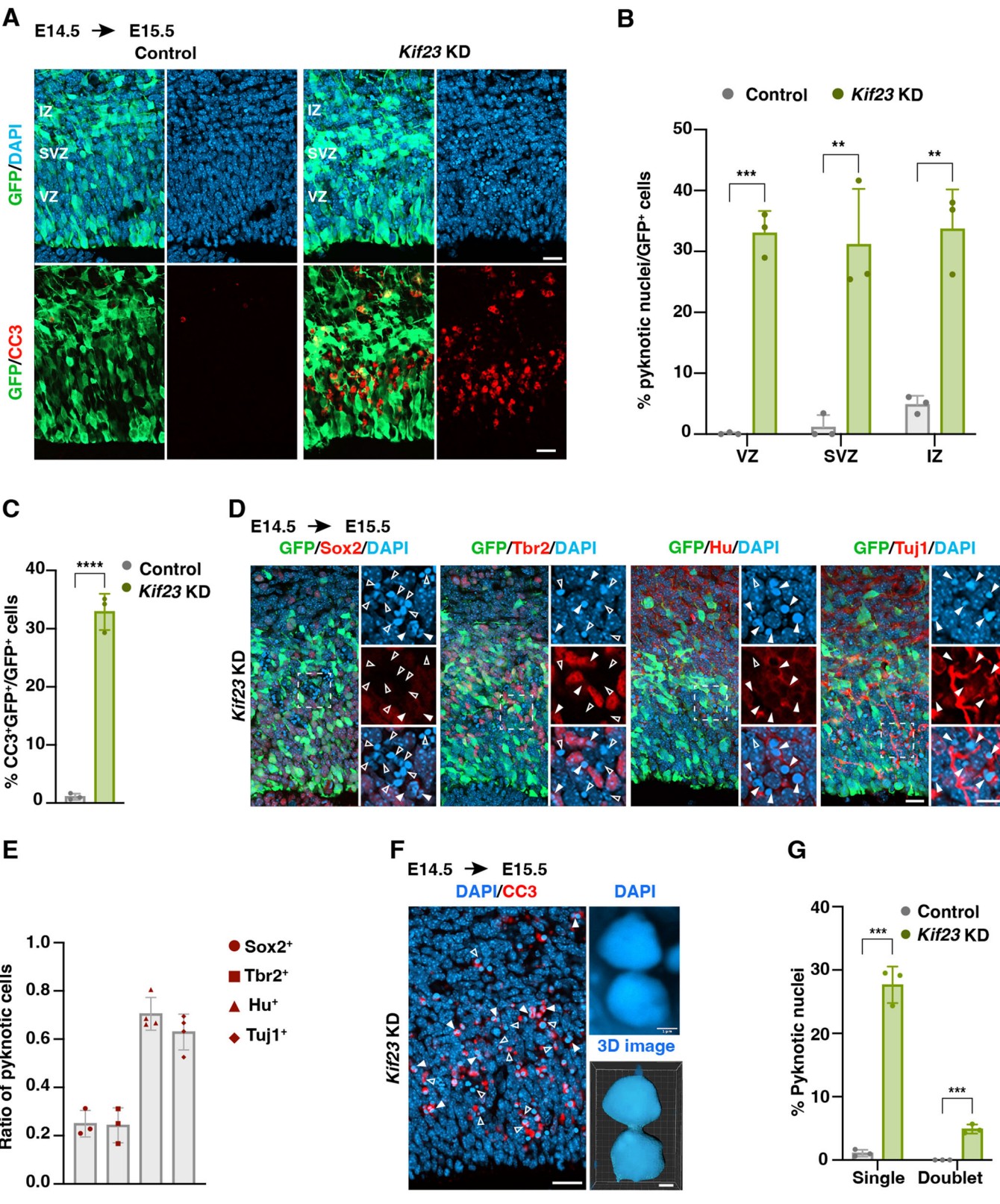

**Figure 3.   Kif23 deficiency leads to cytokinetic defect and neuronal apoptosis.**

(A–C) Representative images of E15.5 control and *Kif23*-KD cortices stained against GFP, CC3, and DAPI showing pyknotic cells across various cortical zones (A). Ventricular zone (VZ), subventricular zone (SVZ), and intermediate zone (IZ). Scale bars, 20 μm. Quantification of the percentage of pyknotic cells VZ, SVZ, and IZ, respectively (B) or CC3$^+$ cells (C) relative to the total GFP$^+$ cells within the 320 μm wide column of E15.5 cortices. The data represent the mean ± SD (Control $n = 1167$ cells, 3 embryos; *Kif23*-KD $n = 1248$ cells, embryos). Multiple *t* tests, *p* values from left to right: ***$p = 0.000294$, **$p = 0.005134$, **$p = 0.003350$ (B). Two-tailed Student's *t* test, ****$p = 6.3E-05$ (C). (D, E) Representative images of E15.5 *Kif23*-KD cortical sections stained for GFP, Sox2, Tbr2, Hu, Tuj1, and DAPI (D). Scale bars, 20 μm. Close arrowheads indicate Sox2$^+$, or Tbr2$^+$, or Hu$^+$, or Tuj1$^+$ pyknotic cells. Open arrowheads indicate Sox2$^-$, or Tbr2$^-$, or Hu$^-$, or Tuj1$^-$ pyknotic cells. Quantification of the ratio of pyknotic cells positive for Sox2, Tbr2, Hu, and Tuj1 within the 110 μm wide column of E15.5 cortices (E). The data represent the mean ± SD ($n = 148$ cells, 3 embryos for Sox2$^+$ group; $n = 131$ cells, 3 embryos for Tbr2$^+$ group; $n = 163$ cells, 4 embryos for Hu$^+$ group; $n = 234$ cells, 4 embryos for Tuj1$^+$ group). (F, G) Representative images of E15.5 *Kif23* siRNA transfected cortices stained for CC3 and DAPI (F). Top right panel: binucleated cell stained with DAPI. Bottom right panel: the 3D image of the binucleated cell. Scale bars, 20 μm (left) and 1 μm (right). Open arrowheads indicate pyknotic single with micronuclei. Closed arrowheads indicate pyknotic doublets. Quantification of the percentage of pyknotic cells relative to the total GFP$^+$ cells within the 320 μm wide column of E15.5 cortices (G). The data represent the mean ± SD (Control $n = 1167$ cells, 3 embryos; *Kif23*-KD $n = 1248$ cells, 3 embryos). Multiple *t* tests, *p* values from left to right: ***$p = 0.000193$, ***$p = 0.000302$. Source data are available online for this figure.

planes decreased (control, 67.19%; *Kif23*-KD, 35.79%) (Fig. 5G,H). These findings collectively underline the impact of Kif23 on the proper orientation of the mitotic spindle, which, when impaired due to Kif23 depletion, may lead to alterations in spindle cleavage plane orientation and, consequently, influence the mode of NSPC division, in line with a previous study (Silver et al, 2010).

## Molecular mechanism of Kif23-mediated cytokinetic regulation in NSPCs

In our quest to unravel the molecular mechanisms by which Kif23 influences cytokinesis in the developing cortex, we further analyzed the localization of ECT2, a Rho guanine nucleotide exchange factor, in dividing NSPCs (Nishimura and Yonemura, 2006). Following *Kif23*-KD, a notable reduction in ECT2 expression within the spindle midzone of dividing NSPCs was observed in comparison to the control group (Fig. 6A,B). Additionally, we explored the localization of RhoA, a key determinant of the division plane (Kamijo et al, 2006) and found a significant decrease in apical RhoA intensity in Kif23-depleted cortices (Fig. 6C,D). Furthermore, we uncovered the co-localization of Kif23 with Cit-K, a RhoA effector kinase, within the midbody (Fig. 6E). Subsequent analysis revealed a notable reduction in the number of Cit-K containing apical midbodies under the *Kif23*-KD condition (Fig. 6F,G). These collective findings strongly underscore the pivotal role of Kif23 in orchestrating the proper localization of critical players like ECT2, RhoA, and Cit-K during the intricate processes of cytokinesis in NSPCs.

## Loss of *Kif23* activates cell death and cell cycle arrest pathways

It is well established that binucleated cells are more susceptible to DNA damage when they re-enter the next cell cycle (Hayashi and Karlseder, 2013; S Pedersen et al, 2016). Given our observation of a substantial presence of binucleated cells, we proceeded to assess DNA damage by staining with the maker γ-H2AX. Notably, within the *Kif23*-KD cortices, we observed a pronounced upregulation of γ-H2AX, which was not evident in the control cortices (Fig. 7A,B). In the realm of NSPCs, DNA damage is known to activate p53, which, in turn, triggers cell cycle arrest and apoptosis (Shimada et al, 2015; Homma et al, 2021). We further probed whether *Kif23*-KD led to the activation of p53. In comparison to the control group,

we noticed a significant increase in the number of active p53$^+$ cells within *Kif23*-KD cortices (Fig. 7C,D).

Given the established role of the cell cycle-dependent kinase inhibitor p21 in promoting progenitor cells to exit the cell cycle and initiate differentiation (Siegenthaler and Miller, 2005), we proceeded to investigate the expression of p21. As anticipated, the results revealed a significant increase in the number of p21$^+$ cells within the *Kif23*-KD cortices when compared to the control (Fig. 7E,F). Consequently, our findings strongly support the notion that Kif23 deficiency not only leads to an increase in apoptosis but also impairs the progression of the cell cycle, consistent with the cell cycle-exit data. Therefore, our findings, in line with previous studies (Minn et al, 1996; Lanni and Jacks, 1998), suggest that Kif23-deficient NSPCs, which experience cytokinetic failure after exiting mitosis and progressing to an interphase state, activate the γ-H2AX-p53-p21 signaling pathway in the next cell cycle to prevent the transition from G1 to S phase. These sequential processes result in precocious exit from the cell cycle and eventually apoptosis of premature neurons.

## Kif23 is essential for maintaining the apical junction structure of NSPCs

Recent research has indicated that adherens junctions remain closely associated with the apical side of the contractile ring during cytokinesis (Nguyen et al, 2014). Microtubules, which constitute the central spindle complex, are implicated in the structural maintenance of adherens junctions in epithelial cells (Meng et al, 2008; Breznau et al, 2015). Moreover, it is known that Kif23 interacts with proteins within the adherens junction (van de Ven et al, 2017). In light of these findings, we next explored to determine whether Kif23 plays a vital role in preserving the integrity of the apical surface structure by observing the distribution of apical surface molecules F-actin and p120-catenin. Our investigations revealed that *Kif23*-KD cortices had a reduced presence of these molecules at the apical surface in comparison to the control group (Fig. 8A). We also observed disruption of the apical structure by immunostaining with the adherens junction protein, β-catenin (Fig. 8A). To gain a comprehensive view of the entire apical surface of NSPCs, we conducted immunostaining on whole mounts of E15.5 *Kif23*-KD cortices using the apical junctional marker ZO-1 (Fig. 8B). Significantly, the size of the apical domain in GFP$^+$ cells exhibited a noteworthy increase under the *Kif23*-KD condition (Fig. 8C,D).

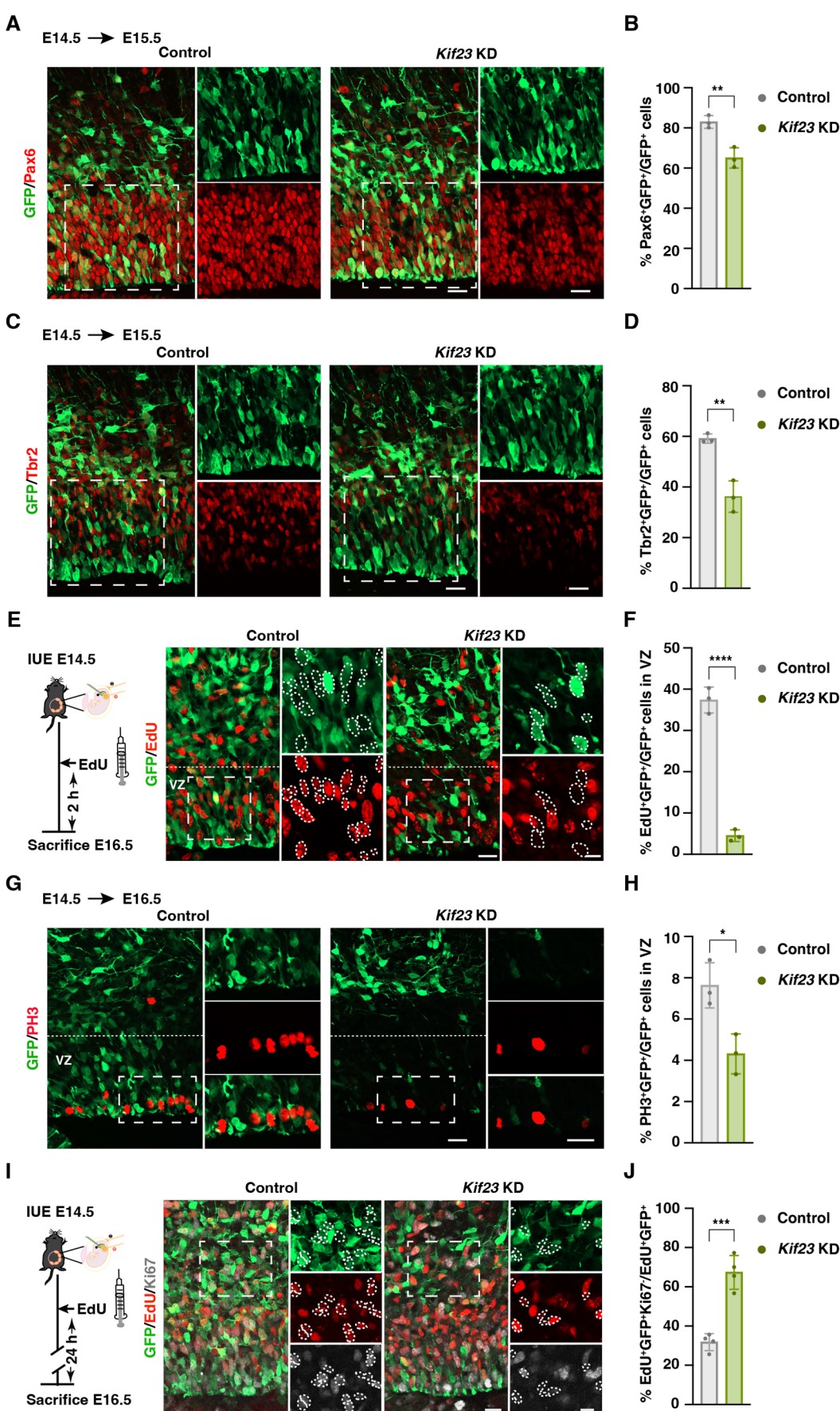

**Figure 4. Kif23 deficiency depletes NSPCs pool and induce premature cell cycle exit.**

(A, B) Representative images of the control and *Kif23*-KD cortices at E15.5 stained for GFP and Pax6 (A). Boxed areas denote zoomed areas. Scale bars, 20 μm. Quantification of the percentage of Pax6$^+$GFP$^+$ cells relative to the total GFP$^+$ cells within the 100 μm wide column of E15.5 cortices (B). The data represent the mean ± SD (Control $n = 524$ cells, 3 embryos; *Kif23*-KD n = 450 cells, 3 embryos). Two-tailed Student's $t$ test, **$p = 0.006541$. (C, D) Representative images of the control and *Kif23*-KD cortices at E15.5 stained for GFP and Tbr2 (C). Boxed areas denote zoomed areas. Scale bars, 20 μm. Quantification of the percentage of Tbr2$^+$GFP$^+$ cells relative to the total GFP$^+$ cells within the 100 μm wide column of E15.5 cortices (D). The data represent the mean ± SD (Control $n = 547$ cells, 3 embryos; *Kif23*-KD $n = 494$ cells, 3 embryos). Two-tailed Student's $t$ test, **$p = 0.003548$. (E, F) Timeline of the in-utero electroporation and EdU injection (left). Electroporated cortical sections were stained for GFP and EdU (right) (E). Dashed lines illustrate the boundary for the ventricular zone (VZ). Boxed areas denote zoomed areas. Scattered lines encircle GFP$^+$ cells that are EdU$^+$ or EdU$^-$. Scale bars, 20 μm. Quantification of the percentage of EdU$^+$GFP$^+$ cells relative to the total GFP$^+$ cells within the VZ of 150 μm wide column of E16.5 cortices (F). The data represent the mean ± SD (Control $n = 415$ cells, 3 embryos; *Kif23*-KD $n = 87$ cells, 3 embryos). Two-tailed Student's $t$ test, ****$p = 7.8E-05$. (G, H) Representative images of the control and *Kif23*-KD cortices at E16.5 stained for GFP and PH3 (G). Dashed lines illustrate the boundary for the ventricular zone (VZ). Boxed areas denote zoomed areas. Scale bars, 25 μm. Quantification of the percentage of PH3$^+$GFP$^+$ cells relative to the total GFP$^+$ cells within the VZ of 200 μm wide column of E16.5 cortices (H). The data represent the mean ± SD (Control $n = 208$ cells, 3 embryos; *Kif23*-KD $n = 72$ cells, 3 embryos). Two-tailed Student's $t$ test, *$p = 0.017095$. (I, J) Timeline of the in-utero electroporation and EdU injection (left). Electroporated cortical sections were stained for GFP, Ki67, and EdU (right) (I). Boxed areas denote zoomed areas. Scattered lines encircle EdU$^+$GFP$^+$ cells that are Ki67$^+$ or Ki67$^-$. Scale bars, 20 μm. Quantification of the percentage of EdU$^+$GFP$^+$Ki67$^-$ cells relative to the total EdU$^+$GFP$^+$ cells within the 200 μm wide column of E16.5 cortices (J). The data represent the mean ± SD (Control $n = 460$ cells, 4 embryos; *Kif23*-KD $n = 121$ cells, 4 embryos). Two-tailed Student's $t$ test, ***$p = 0.000318$. Source data are available online for this figure.

Our analysis of nuclear morphology using DAPI staining in whole mounts of E15.5 *Kif23*-KD cortices showed no pyknotic doublets on the apical surface (Appendix Fig. S4A), suggesting that the increase in apical domain size could be caused by compromised integrity of apical junctions rather than the presence of pyknotic doublets. The centrosome in NSPCs, a crucial component of the apical membrane, is anchored by interactions with microtubules and junctions within the apical domain (Yang et al, 2021). To address the points, we further quantified the number of apical centrosomes by immunostaining and observed a significant reduction in the number of γ-tubulin positive centrosomes following *Kif23*-KD (Appendix Fig. S4B,C). These findings consistently suggest that Kif23 potentially plays a crucial role in maintaining the integrity of apical junctions in NSPCs.

## Microcephaly-related *KIF23* mutation fails to rescue *Kif23*-KD phenotypes

Since mutations of the human *KIF23* gene are reported in cases of microcephaly, we explored the role of KIF23. We first confirmed the expression of *KIF23* using scRNA-seq datasets obtained from human fetal cortex during mid-gestation (specifically, gestation week 17-18) (Polioudakis et al, 2019; Data Ref: Polioudakis et al, 2019). Our analysis revealed that *KIF23* was notably enriched within the cycling progenitor clusters, specifically those at the S (PgS) and G2/M (PgG2M) phases (Fig. 9A). Furthermore, we assessed the expression of *Kif23* using the scRNA-seq datasets from the cortical germinal zones of the developing ferret at embryonic day 34 and postnatal day 1 (Del-Valle-Anton et al, 2024; Data Ref: Del-Valle-Anton et al, 2024). We found that in the ferret cortex, *Kif23* is expressed in NSPC (i.e., radial glial cells, RGCs) and IP clusters in the S/G2/M phases (Appendix Fig. S5). The cells expressing *KIF23*/*Kif23* in the human fetal cortex and the ferret fetal and postnatal cortices exhibited features similar to the mouse cells expressing *Kif23* (Fig. 1D; Appendix Fig. S1A,B), underscoring the conservation of Kif23 function in the developing cortex across mammals, including mice, ferrets, and humans.

Next, we asked about potential means of a mutation related to microcephaly by rescuing *Kif23*-deficient condition. We employed human *KIF23* (*hKIF23*) and a variant carrying a missense mutation within the motor domain of *KIF23* (c.755 T > A; p.L266H) (referred to as *mhKIF23*). This specific mutation has been identified in patients with microcephaly in a homozygous condition (Karaca et al, 2015). We first electroporated either mock *pCAX* or *hKIF23* into mouse embryos and confirmed the expression of hKIF23 (Fig. EV3A,B). To assess the rescue potential, we co-electroporated *hKIF23* along with *Kif23*-siRNA into mouse embryos at E14.5 and collected the brains at E15.5 to perform immunostaining. The outcome revealed that *Kif23* siRNA effectively reduced the amount of mouse Kif23 protein, while human KIF23 protein remained unaffected (Fig. EV3A,C). Furthermore, we found that *mhKIF23* also exhibited resistance to *Kif23*-siRNA when co-electroporated along with *Kif23*-siRNA (Fig. EV3A,C).

Subsequently, we investigated the distributions of GFP$^+$ cells at E16.5 in brains co-electroporated with *hKIF23* and *Kif23*-siRNA at E14.5. Under this condition, the distribution of GFP$^+$ cells closely resembled that of the control group, indicating the successful rescue of *Kif23*-KD by *hKIF23* (Fig. 9B–D). In contrast, the co-electroporation of *mhKIF23* and *Kif23*-siRNA resulted in a decrease in the proportion of GFP$^+$ cells within the VZ/SVZ and an increase in the IZ, as compared to the control and *hKIF23* groups (Fig. 9B–D). These results provide a strong indication that *mhKIF23* was unable to rescue the defects in cortical development induced by *Kif23*-KD. Since co-electroporation of *mhKIF23* showed a statistically significant difference in distribution of GFP$^+$ cells compared to the *Kif23*-KD group, it is suggested that *mhKIF23* may retain a partial activity. This observation strongly implies that the mutation within the *hKIF23* gene can result in a partial loss of function, which, in turn, can lead to the onset of microcephaly, thus furthering our understanding of the genetic basis of this neurodevelopmental disorder.

## Discussion

In this study, we delved into the dynamic localization and developmental transition of the motor protein Kif23 in NSPCs and its critical roles in multiple processes of mitosis during early corticogenesis in mice. Our research uncovered that Kif23 exhibited dynamic localization patterns that change as NSPCs progress through different phases of the cell cycle. Notably, it was consistently present in the spindle microtubule throughout the mitotic stages of NSPCs.

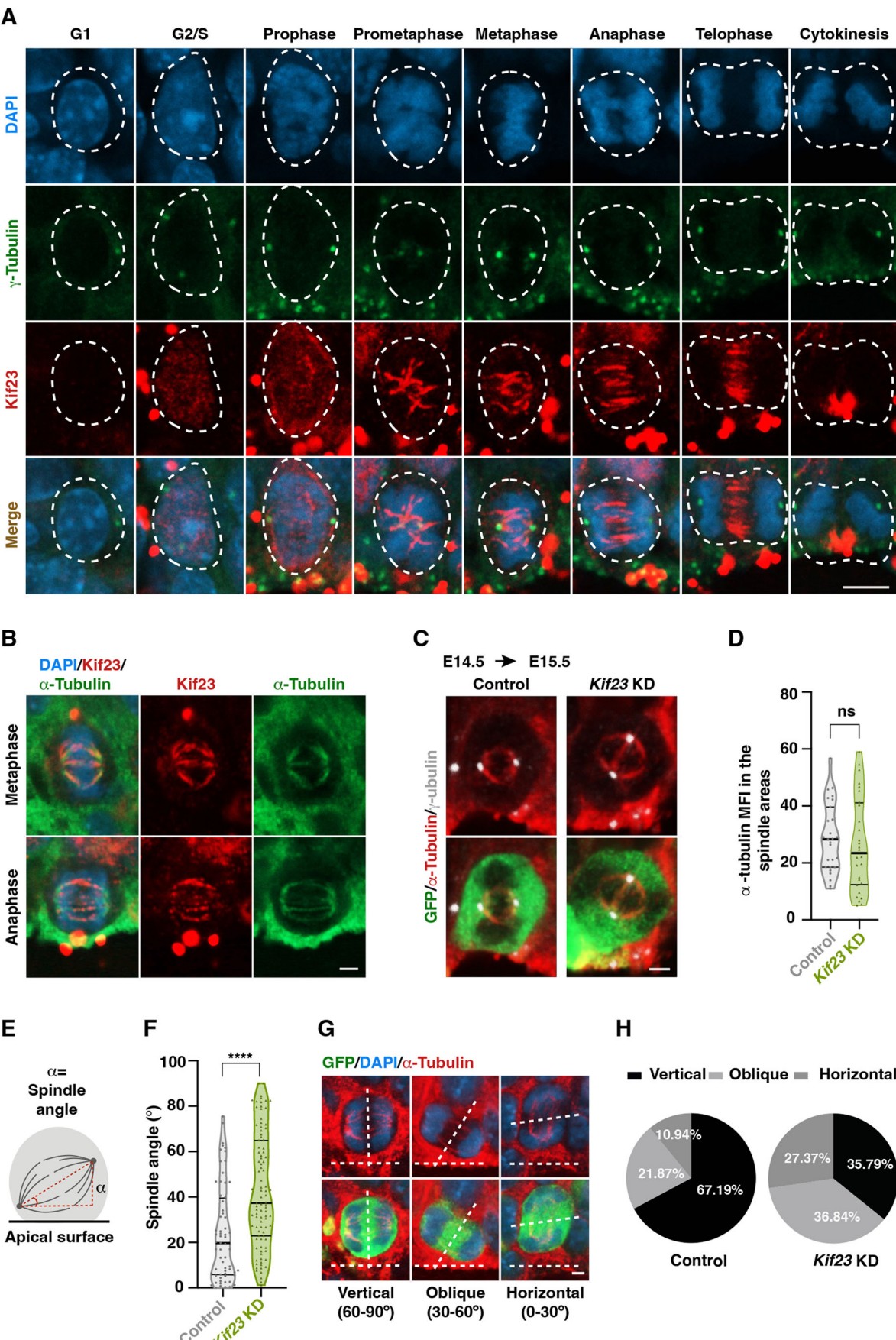

◀ **Figure 5. Knockdown of *Kif23* disrupts the orientation of mitotic spindle.**

(A, B) Representative images of WT E14.5 cortex stained for Kif23, DAPI and γ-tubulin (centrosomal marker) (A) or α-tubulin (microtubule marker) (B). Dashed lines indicate cells in the different stages of the cell cycle. Scale bars, 5 μm (A); 2 μm (B). (C, D) Representative images of the electroporated brain sections at E15.5 stained for GFP, α-tubulin, and γ-tubulin (C). Scale bar, 2 μm. Quantification of the mean fluorescence intensity of α-tubulin in the spindle areas of GFP⁺ metaphase cells (D). The data represent *n* = 27 cells, 9 embryos for the control group; *n* = 28 cells, 11 embryos for the *Kif23*-KD group. The thick and thin black horizontal lines represent the medians and the quartiles, respectively. Two-tailed Student's *t* test, ns: not significant. (E, F) Graphical representation of spindle angle measurement (E). Distribution of spindle angles (°) (F). The data represent *n* = 64 cells, 10 embryos for the control group; n = 95 cells, 12 embryos for the *Kif23*-KD group. The thick and thin black horizontal lines represent the medians and the quartiles, respectively. Two-tailed Student's *t* test, ****p = 1.8E-06. (G, H) Representative images of vertical, oblique, and horizontal cleavage plane orientation stained against GFP, α-tubulin, and DAPI (G). Scale bar, 2 μm. White dashed lines indicate the cleavage plane. Quantification of the percentage of each class of spindle cleavage plane orientation (H). Data represent *n* = 64 cells, 10 embryos for the control group; *n* = 95, 12 embryos for the *Kif23*-KD group. Source data are available online for this figure.

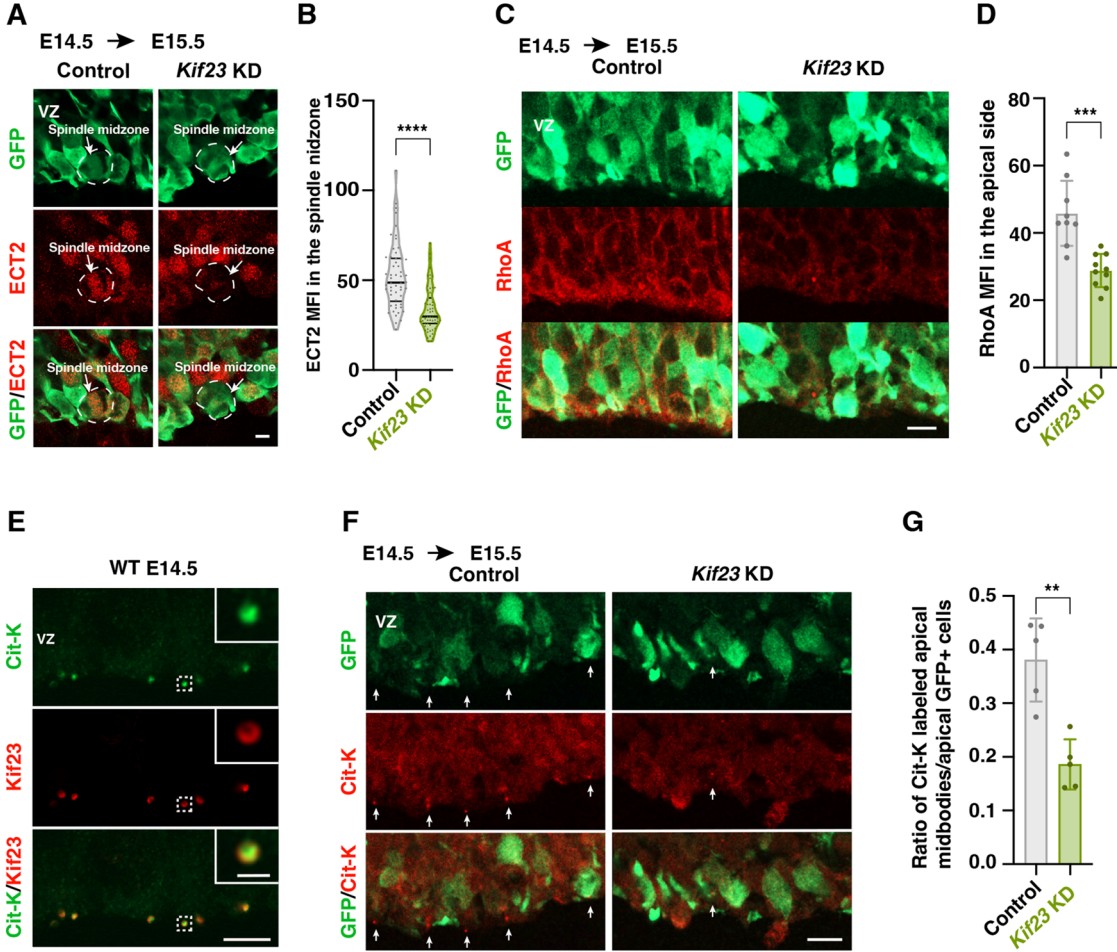

**Figure 6. Kif23 deficiency impairs the localization of cytokinetic molecules.**

(A, B) Images of E15.5 control and *Kif23*-KD cortical sections stained for GFP and ECT2 (A). Examples of cells at the anaphase stage are outlined in the ventricular zone (VZ), and arrows point to the spindle midzone. Scale bar, 5 μm. Quantification of the mean fluorescence intensity of ECT2 in the spindle midzone of GFP⁺ anaphase cells (B). The data represent *n* = 54 cells, 11 embryos for the control group; *n* = 58 cells, 16 embryos for the *Kif23*-KD group. The thick and thin black horizontal lines represent the medians and the quartiles, respectively. Two-tailed Student's *t* test, ****p = 5.8E-09. (C, D) Images of E15.5 control and *Kif23*-KD cortical sections stained for GFP and RhoA to show RhoA localization in the VZ containing GFP⁺ cells (C). Scale bar, 10 μm. Quantification of the mean fluorescence intensity of apical RhoA (within 30 μm from the GFP⁺ ventricular surface) (D). The data represent the mean ± SD (Control *n* = 9 embryos; *Kif23*-KD *n* = 10 embryos). Two-tailed Student's *t* test, ***p = 0.000129. (E) Images of WT E14.5 cortex to show the co-localization of Cit-K and Kif23 at the apical surface of the VZ. Scale bars, 2 μm. (F, G) Images of E15.5 control and *Kif23*-KD cortical sections stained for GFP and Cit-K (F). Arrows point to apical midbodies in the VZ. Scale bar, 10 μm. Quantification of the ratio of Cit-K labeled apical midbodies relative to the apical GFP⁺ cells within the 160 μm wide column of E15.5 cortices (G). The data represent the mean ± SD (Control *n* = 158, 5 embryos; *Kif23*-KD *n* = 173, 5 embryos). Two-tailed Student's *t* test, **p = 0.001335. Source data are available online for this figure.

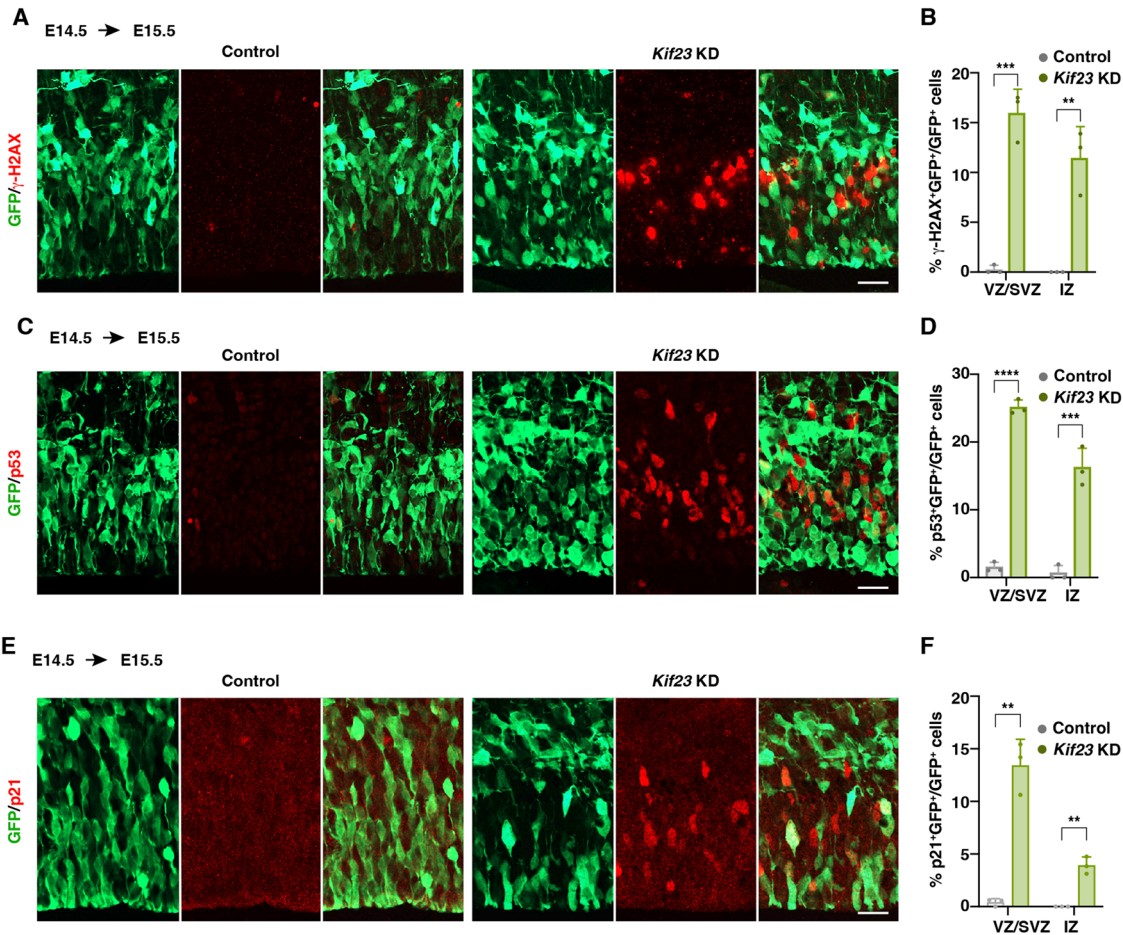

**Figure 7.  Knockdown of *Kif23* results in upregulation of cell cycle arrest and cell apoptosis pathway.**

(A–F) Representative images of E15.5 control and *Kif23*-KD cortices stained for GFP and γ-H2AX (A) or GFP and p53 (C) or GFP and p21 (E). Scale bars, 20 μm. Quantification of the percentage of GFP⁺ cells that are γ-H2AX⁺ (B), p53⁺ (D) or p21⁺ (F). The data represent the mean ± SD (Control $n=510$ cells, 3 embryos; *Kif23*-KD $n=551$ cells, 3 embryos for γ-H2AX; Control $n=1054$ cells, 3 embryos; *Kif23*-KD $n=1257$ cells, 3 embryos for p53; Control $n=650$ cells, 3 embryos; *Kif23*-KD $n=574$ cells, 3 embryos for p21). Multiple *t* tests, *p* values from left to right: ***$p=0.000861$, **$p=0.007526$ (B); ****$p=1.3E\text{-}05$, ***$p=0.000936$ (D); **$p=0.001724$, **$p=0.001724$ (F). Source data are available online for this figure.

As a general principle, the orientation of the spindle cleavage planes has far-reaching consequences in determining the mode of cell division. Spindle cleavage planes oriented vertically in relation to the apical surface tend to yield symmetric division, generating two proliferative progenitors, while horizontally or obliquely oriented spindle cleavage planes result in asymmetric division, producing one proliferative daughter cell and one postmitotic neuron (Haydar et al, 2003; Noctor et al, 2004). Our findings demonstrated that the disruption of Kif23 function in the developing cortex led to abnormalities in spindle orientation, manifesting as an increased prevalence of NSPCs with asymmetric spindle cleavage planes. These observations are in line with prior studies implicating mutations in several centrosomal genes like *Nde1*, *Cdk5rap2*, and *PPP4c*, which similarly induce spindle orientation defects, resulting in an increase in asymmetric mitotic cleavage and the premature exit of NSPCs from the cell cycle (Feng and Walsh, 2004; Lizarraga et al, 2010; Xie et al, 2013). The defect in the spindle cleavage plane orientation might affect the maintenance and production of NSPCs and neurons by changing

the distribution of cell fate determinants (Chenn and McConnell, 1995; Zhong et al, 1996). Therefore, our study postulated that the precocious neurogenesis observed by *Kif23*-KD, is likely mediated by spindle orientation defects.

Improper cytokinesis can manifest in two distinct outcomes: (1) binucleation, which arises from cleavage furrow regression, and (2) the formation of individual cells with micronuclei, a consequence of premature chromatin bridge resolution (Gromley et al, 2003; Pampalona et al, 2012). Our research underscores that deficiencies in Kif23 function precipitated both of the phenotypes: the presence of doublet pyknotic cells and single pyknotic cells with micronuclei, ultimately culminating in reduced neuronal production. Consequently, our findings suggest the indispensable role of Kif23, which extends beyond its involvement in regulating the mitotic spindle to encompass the pivotal task of ensuring proper cytokinesis.

Our investigation unveiled that *Kif23*-KD had a notable impact on the expression of cytokinetic molecules within NSPCs, including ECT2, RhoA, and Cit-k. Recent research in *Drosophila* has shed light on the necessity of Kif23 for the transportation of ECT2 as

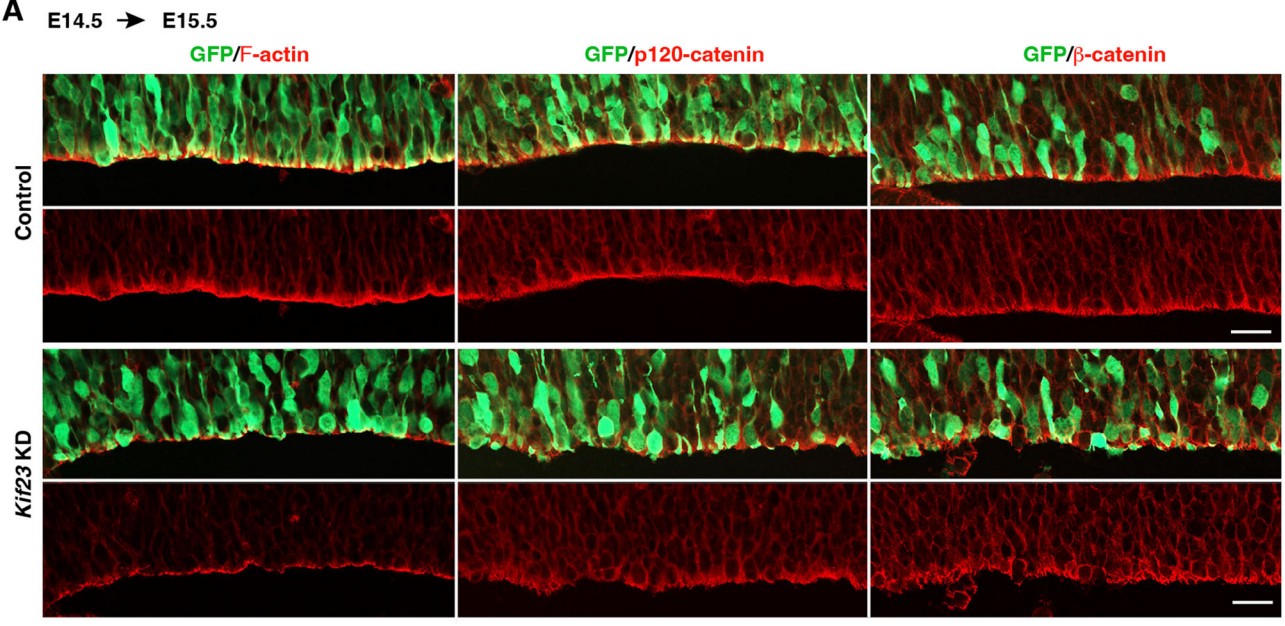

A

E14.5 → E15.5

GFP/F-actin    GFP/p120-catenin    GFP/β-catenin

Control

Kif23 KD

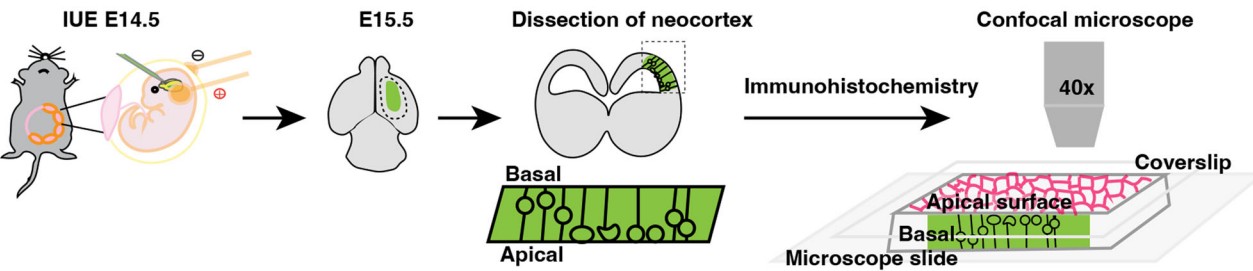

B

IUE E14.5    E15.5    Dissection of neocortex    Confocal microscope

40x

Immunohistochemistry

Basal

Apical

Apical surface

Basal

Coverslip

Microscope slide

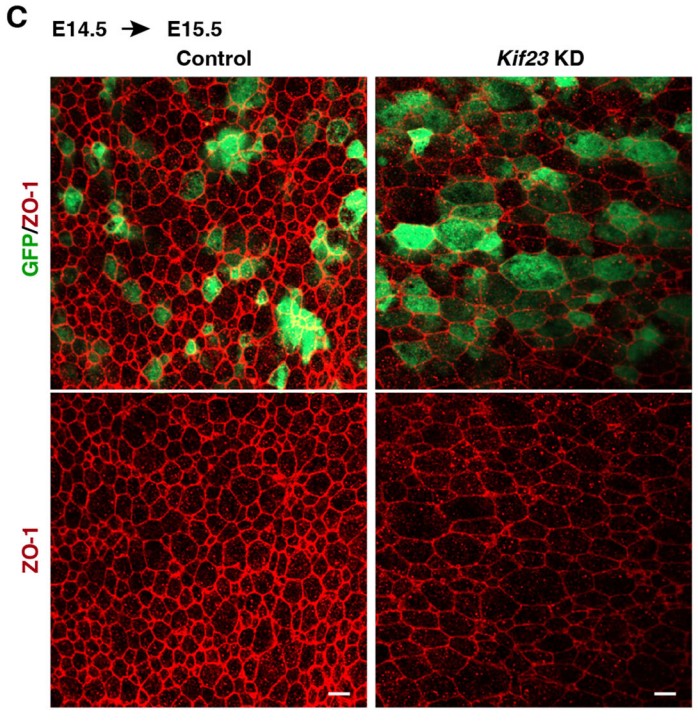

C

E14.5 → E15.5

Control    Kif23 KD

GFP/ZO-1

ZO-1

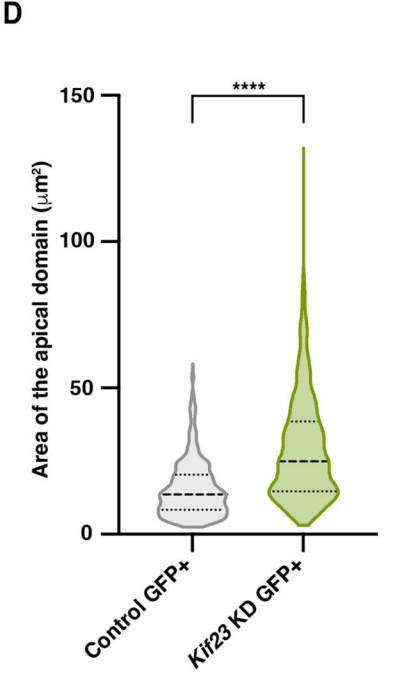

D

****

Area of the apical domain (μm²)

Control GFP+    Kif23 KD GFP+

◀  **Figure 8.    Knockdown of *Kif23* results in the disruption of apical junction integrity.**

(A) Representative images of the control and *Kif23*-KD cortical sections at E15.5 stained for GFP, F-actin, p120-catenin, and β-catenin. Scale bars, 20 μm. (B–D) Schematic illustration of the whole mount immunohistochemistry to observe the apical structure (B). Whole-mount images of the electroporated brain sections immunostained for GFP and ZO-1 (C). Scale bars, 20 μm. Quantification of the apical domain areas (D). The data represent $n = 363$ apical domains, 4 embryos for the control group; $n = 475$ apical domains, 6 embryos for the *Kif23*-KD group. The thick and thin black horizontal dash lines represent the medians and the quartiles, respectively. Two-tailed Student's *t* test, ****$p < 1.0E\text{-}15$. Source data are available online for this figure.

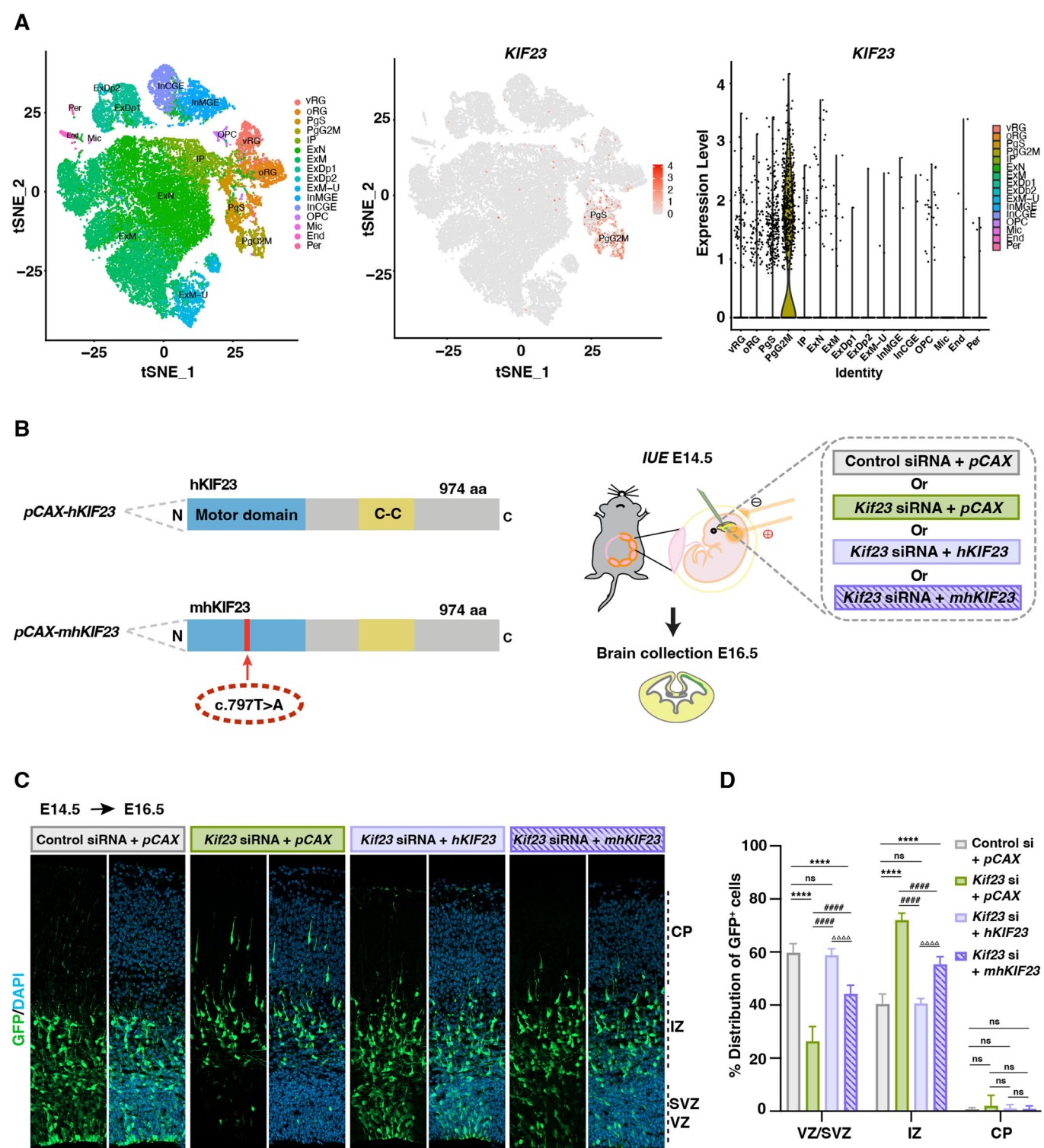

◀ **Figure 9.**  *Kif23* knockdown defects can be rescued by WT human KI23 but not by mutant found in microcephaly patients.

(A) Single-cell RNA-seq analysis of gestational week 17–18 human fetal cortex showing cell type clusters (left), feature plots (middle), and violin plots (right) to show the expression of *KIF23*. The dataset was obtained from Polioudakis et al, 2019. vRG ventricular radial glia, oRG outer radial glia, PgS progenitors in S phase, PgG2M progenitors in G2M phase, IP intermediate progenitor, ExN excitatory newborn neurons, ExM maturing excitatory neurons, ExM-U maturing upper layer excitatory neurons, ExDp deep layer excitatory neurons, InMGE medial ganglionic eminence interneurons, InCGE caudal ganglionic eminence interneurons, OPCs oligodendrocyte precursor cells, End endothelial cells, Per pericytes. (B–D) Diagram of WT human *KIF23* and mutant *KIF23* constructs, and overview of the experiment (B). Representative images of the mouse cortices two days after in-utero electroporation of indicated constructs stained for GFP and DAPI (C). Scale bar, 50 μm. Quantification of the distribution of GFP⁺ cells in VZ/SVZ, IZ, and CP, respectively within the 200 μm wide column of E16.5 cortices (D). The data represent the mean ± SD (n = 1379 cells, 6 embryos for control siRNA + *pCAX* group; n = 507 cells, 5 embryos for *Kif23* siRNA + *pCAX* group; n = 768 cells, 4 embryos for *Kif23* siRNA + *pCAX-hKIF23* group; n = 1467 cells, 8 embryos for *Kif23* siRNA + *pCAX-mhKIF23* group). One-way ANOVA with Bonferroni's multiple comparisons test, *p* values from left to right: \*\*\*\**p* = 4.5E-11, \*\*\*\**p* = 1.9E-06, \*\*\*\**p* = 1.6E-12, \*\*\*\**p* = 7.4E-08, ns: not significant (vs control siRNA + *pCAX* group); ####*p* = 4.2E-10, ####*p* = 4.9E-07, ####*p* = 1.2E-11, ####*p* = 3.5E-08, ns: not significant (vs *Kif23* siRNA + *pCAX* group); ^△△△△*p* = 2.4E-05, ^△△△△*p* = 7.8E-07, ns: not significant (vs *Kif23* siRNA + *pCAX-mhKIF23* group). Source data are available online for this figure.

cargo to the spindle midzone and the equatorial cell cortex (Warecki and Tao, 2023). However, to comprehensively understand the underlying mechanisms, further study is warranted to ascertain whether Kif23 physically interacts with ECT2, RhoA, and Cit-k within NSPCs. Additionally, investigating whether these molecules collaborate in tandem or operate independently in the regulation of cytokinesis represents an intriguing avenue for future exploration.

The molecular mechanisms underlying the detection of cytokinetic abnormalities and initiation of cell apoptosis in the developing brain are not well understood. Previous studies have indicated that when cells are arrested in the G1 phase, they tend to exit the cell cycle rather than undergo apoptosis, both in cultured cells and in the fly brain (Ganem et al, 2014; Gogendeau et al, 2015). In our study, we identified an augmented DNA damage response and increased p53 signaling in the *Kif23*-KD cortices. Our findings corroborate a previous study indicating p53-dependent apoptosis was also observed in *Kif20b* mutant cortices, although it was less severe than in our *Kif23*-KD cortices because no DNA damage response was found (Little and Dwyer, 2019). Notably, our investigation suggests that Kif23 exerts a more prominent influence on cell survival regulation compared to other members of Kinesin-6 family. Additionally, we observed p53 accumulation in both NSPCs and neurons, alongside elevated p21 levels after *Kif23*-KD. This elevated p53 and p21 may account for the premature neuronal differentiation observed in *Kif23*-KD cortices. Since active p53 was detected in both NSPCs and neurons following *Kif23*-KD, it suggests that p53 accumulation initiates in NSPCs due to cytokinesis failure in response to DNA damage and persists in the differentiated neurons, ultimately triggering neuronal apoptosis. Therefore, our findings propose a mechanism linking cytokinesis failure, premature cell cycle exit, and subsequent apoptosis induced by Kif23 dysfunction.

We found a reduction in Tbr2⁺ IPs as seen in the RGCs in the *Kif23*-KD cortices. Based on our observation that Kif23 deficiency primarily impaired mitotic activity in RGCs, but not in IPs, the decrease of IPs in *Kif23*-KD cortices is likely due to a failure to generate IPs from RGCs. Additionally, Kif23 was predominantly expressed in a large population of RGCs compared to the population of IPs, and few pyknotic nuclei were observed in both RGCs and IPs after *Kif23*-KD. It is thus suggested that Kif23 may primarily function in RGCs, although it is difficult to exclude the possibility that *Kif23*-KD may directly regulate apoptosis in a subset of the IP population, given the expression of Kif23 in dividing IPs. In regard with brain evolution, the expansion of the

cortical surface in gyrencephalic species is correlated with an increase in the SVZ due to an expansion in the IP population (Martínez-Cerdeño et al, 2006; Striedter and Charvet, 2009). Interestingly, we noticed in the public scRNA-seq datasets of the developing gyrencephalic cortex of the ferret that *Kif23* is expressed in multiple IP clusters along with RGCs (Del-Valle-Anton et al, 2024). It is thus possible to assume that Kif23 may have a significant role in NSPCs/IPs within the mammalian gyrencephalic neocortex, a topic of considerable interest.

Kif23 is found in the midbodies located on the apical surface of the embryonic mouse cortex (Maliga et al, 2013). Symmetrically dividing NSPCs release their midbodies into the ventricular fluid after cytokinesis (Dubreuil et al, 2007), as observed in our samples stained with Kif23 (Fig. 1E). Kif23 expression was notably higher during early neurogenesis, a phase when NSPCs predominantly undergo symmetric division, suggesting a possible association between Kif23-containing midbody release and symmetric division (Fig. 1E). In HeLa cells, Kif23 facilitates the localization of midbody-enriched mRNA cargo, leading to translational availability after the M/G1 transition, with implications for cell fate determination, cell division, and proliferation (Park et al, 2023). This raises the intriguing possibility that Kif23, functioning as a motor protein, might transport cell fate determinants within the NSPCs, influencing the fate of their daughter cells.

We observed a significant deterioration in the structure of apical endfeet in *Kif23*-KD cortices, highlighting another crucial role of Kif23 in supporting NSPCs. A study in *C. elegans* revealed that the dynamic localization of the adhesion complex during cytokinesis plays a pivotal role in preserving proper cell contacts and maintaining the integrity of apical surface architectures (Bai et al, 2020). Considering Kif23's orientation towards microtubule plus ends, its role in maintaining the connection between microtubule and junctional proteins becomes pivotal, especially since microtubule plus ends terminate at adherens junctions during cytokinesis (Shahbazi et al, 2013). It is conceivable that the loss of Kif23 may lead to a reduction in the expression of junctional proteins such as F-actin and p120-catenin within the NSPCs. Importantly, disruptions of the apical structure of NSPCs are well-documented to trigger premature neuronal differentiation (Tamai et al, 2007; Shinohara et al, 2013; Yoon et al, 2017). Furthermore, the number of apical centrosomes was profoundly decreased in the *Kif23*-KD cortices. Along with the disruption of apical integrity, the loss of apical centrosomes and increase in the abventricular centrosomes are often considered a sign of NSPC delamination (Shinohara et al,

																	

2013; Tavano et al, 2018; Uzquiano et al, 2019; Lin et al, 2020). Although we did not see any noticeable increase in abventricular centrosomes, it is possible that delaminated NSPCs might already be differentiated into neurons. Another possibility is that *Kif23*-KD might not directly induce NSPC delamination, but lead to the loss of apical centrosomes by reducing the number of NSPCs.

Microcephaly can stem from impaired neurogenesis, disrupting the delicate balance between NSPC proliferation and neuronal differentiation (Buchman et al, 2010; Chen et al, 2014). In various microcephaly mutants, the presence of apoptotic progeny is frequently noted alongside precocious neuronal production (McMahon et al, 2014; Mao et al, 2015). Notably, cerebral organoids using iPS cells derived from microcephaly patients often manifest premature neuronal differentiation at the expense of progenitor depletion (Lancaster et al, 2013; Zhang et al, 2019). Leveraging a public transcriptomic database of the human fetal cerebral cortex, we identified robust expression of *KIF23* in dividing NSPCs (Polioudakis et al, 2019), suggesting its potential role in human brain development. In the present study, we assessed the human KIF23 function through a *Kif23*-KD model in the mouse cortex. Our findings revealed that WT *hKIF23* effectively rescued the phenotypes of *Kif23*-KD, whereas a motor domain mutant construct (*mhKIF23*) resulted in a partial rescue. Considering that this is a missense mutation, and that severe microcephaly has only been identified in individuals who are homozygous for this mutation (Karaca et al, 2015), it is plausible that this missense mutation might partially impair KIF23 function rather than completely abolishing it. The motor domain of kinesins is responsible for microtubule binding and microtubule-dependent ATPase activity (Verhey et al, 1998), and mutations within the

motor domain are often known to impair the enzymatic and binding activities, resulting in loss of microtubule motility (Nakata and Hirokawa, 1995). Since most mutations in the core motor domain are predicted to affect motility (Gabrych et al, 2019; Budaitis et al, 2021), further investigation into how the mutation within the motor domain of *hKIF23* affects its activity might yield new insights into the mechanistic understanding of the mutation in the pathology of microcephaly.

In summary, our findings underscore the critical importance of proper motor protein function in multiple aspects of NSPC division and the preservation of their apical structure in the early development of the mammalian cortex. Further investigations encompassing human genetics and animal models promise to unveil the intricate molecular and cellular mechanisms underpinning the fascinating process of brain development.

# Methods

## Methods and protocols

### Data processing of Visium spatial gene expression profiles

Spatial transcriptome analysis (Visium, 10x Genomics) was performed on the brain sample of WT E15.5 mice as described previously (Tsai et al, 2024). Briefly, sagittal brain sections (10-μm thick) were placed on gene expression (GE) slides (10X Genomics). The captured cDNA library tagged with spatial barcode were sequenced, and the resulting data was analyzed by Space Ranger software (Version 1.2.1). Gene alignment was

**Reagents and tools table**

| Reagent/resource | Reference or source | Identifier or catalog number |
|---|---|---|
| **Experimental models** | | |
| Mouse: C57BL/6J | CLEA Japan | N/A |
| **Recombinant DNA** | | |
| *pCAG-EGFP* | Gift from Prof. Tetsuichiro Saito, Chiba University, Japan | N/A |
| *pCAX-hKIF23* | This study | N/A |
| *pCAX-mhKIF23* | This study | N/A |
| **Antibodies** | | |
| Chicken anti-GFP (1:1000) | Abcam | Cat# ab13970, RRID:AB_300798 |
| Rabbit anti-Pax6 (1:1000) | MBL International | Cat# PD022, RRID:AB_1520876 |
| Rabbit anti-Tbr2 (1:1000) | Abcam | Cat# ab23345, RRID:AB_778267 |
| Rabbit anti-Ki67 (1:500) | Abcam | Cat# ab15580, RRID:AB_443209 |
| Rabbit anti-Kif23 (1:1000) | Invitrogen | Cat# MA524855 RRID:AB_2717293 |
| Rabbit anti-active p53 (1:1000) | Leica Biosystems | Cat# P53-CM5P, RRID:AB_2744683 |
| Rabbit anti-active caspase-3 (1:1000) | BD Biosciences | Cat# 559565, RRID:AB_397274 |
| Rabbit anti-ECT2 (1:1000) | Millipore | Cat# 07-1364, RRID:AB_10805932 |
| Rabbit anti-γ-tubulin (1:1000) | Abcam | Cat# ab11317, RRID:AB_297921 |
| Rabbit anti-PH3 (1:1000) | Millipore | Cat# 06-570, RRID:AB_310177 |
| Rabbit anti-NeuroD2 (1:1000) | Abcam | Cat# ab104430, RRID:AB_10975628 |

| Reagent/resource | Reference or source | Identifier or catalog number |
|---|---|---|
| Rat anti-Tbr2 (1:1000) | Invitrogen | Cat# 14487582<br>RRID:AB_11042577 |
| Goat anti-Sox2 (1:500) | R and D Systems | Cat# AF2018, RRID:AB_355110 |
| Mouse anti-active caspase-3 (1:1000) | Invitrogen | Cat# 43-7800, RRID:AB_2533540 |
| Mouse anti-β catenin (1:1000) | BD Biosciences | Cat# 610153, RRID:AB_397554 |
| Mouse anti-p120 catenin (1:1000) | BD Biosciences | Cat# 610133, RRID:AB_397536 |
| Mouse anti-p21(1:1000) | Proteintech Group | Cat# 28248-1-AP,<br>AB_2881097 |
| Mouse anti-PH3 (1:1000) | Cell Signaling | Cat# 9706, RRID:AB_331748 |
| Mouse anti-Tuj1(1:1000) | BioLegend | Cat# 801201,<br>RRID AB_2313773 |
| Mouse anti- HuC/D (1:200) | Molecular Probes | Cat# A21271, RRID:AB_221448 |
| Mouse anti-α-tubulin (1:200) | Sigma-Aldrich | Cat# T6199, RRID:AB_477583 |
| Mouse anti-γ-tubulin (1:300) | Sigma-Aldrich | Cat# T6557, RRID:AB_477584 |
| Mouse anti-γ-H2AX (1:1000) | Millipore | Cat# 05-636, RRID:AB_309864 |
| Mouse anti-Cit-k (1:1000) | BD Biosciences | Cat# 611376, RRID:AB_398898 |
| Mouse anti-RhoA (1:100) | Santa Cruz | Cat# sc-418, RRID:AB_628218 |
| Donkey anti-rabbit IgG, Cy3-conjugated (1:500) | Jackson Immunoresearch Laboratories | Cat# 711-165-152,<br>RRID:AB_2307443 |
| Donkey anti-rat IgG, Cy3-conjugated (1:500) | Jackson Immunoresearch Laboratories | Cat# 712-165-153,<br>RRID:AB_2340667 |
| Donkey anti-mouse IgG, Cy3-conjugated (1:500) | Jackson Immunoresearch Laboratories | Cat# 715-165-151,<br>RRID:AB_2315777 |
| Donkey anti-mouse IgG, Alexa 488-conjugated (1:500) | Jackson Immunoresearch Laboratories | Cat# 715-545-150,<br>RRID:AB_2340846 |
| Donkey anti-goat IgG, Alexa 488-conjugated (1:500) | Jackson Immunoresearch Laboratories | Cat# 705-545-147,<br>RRID:AB_2336933 |
| Donkey anti-chicken IgY, Alexa 488-conjugated (1:500) | Jackson Immunoresearch Laboratories | Cat# 703-545-155,<br>RRID:AB_2340375 |
| Donkey anti-rabbit IgG, Alexa 647-conjugated (1:500) | Jackson Immunoresearch Laboratories | Cat# 711-605-152,<br>RRID:AB_2492288 |
| Donkey anti-mouse IgG, Alexa 647-conjugated (1:500) | Jackson Immunoresearch Laboratories | Cat# 715-605-151,<br>RRID:AB_2340863 |
| **Oligonucleotides and other sequence-based reagents** | | |
| Scramble control RNAi<br>CAGAUGCGUGACGGCAGAACCAAUU | Invitrogen | https://rnaidesigner.thermofisher.com/rnaiexpress/ |
| Kif23 RNAi<br>CAGACUACGUGAGGCCGGAAACAUU | Invitrogen | https://rnaidesigner.thermofisher.com/rnaiexpress/ |
| Primers for Kif23 cDNA amplification for *Kif23* probe synthesis (forward: CGGTATCGATAAGCTTATATGATCTATTGGAAGAAGTGC; reverse: TAGAACTAGTGGATCAATAGATGGGTTAGCTTTGAATCTC | This study | N/A |
| Synthetic hKIF23 ORF | eurofins | NM_001367805 |
| Primers to amplify hKIF23 ORF for cloning into the *pCAX* vector (forward: TTTTGGCAAAGAATTGCCACCATGAAGTCAGCG; reverse: TAGCTGGCCAGGATCTCATGGCTTTTTTGCGCTTG | This study | N/A |
| **Chemicals, enzymes and other reagents** | | |
| RNeasy Plus Mini Kit | QIAGEN | Cat# 74134 |
| SuperScipt™III First-Strand Synthesis System | Invitrogen | Cat# 18080400 |
| In-Fusion system | TaKaRa | Cat# Z9648N |
| KOD plus DNA polymerase | TOYOBO | Cat# KOD-201 |
| DIG RNA labeling kit | Roche | Cat# 11277073910 |

| Reagent/resource | Reference or source | Identifier or catalog number |
|---|---|---|
| Bovine serum albumin | Sigma-Aldrich | Cat# A2153-50G |
| Paraformaldehyde | Nacalai Tesque | Cat# 26126-25 |
| 4′,6-Diamidino-2-phenylindole dihydrochloride | Sigma-Aldrich | Cat# D9542-1MG |
| Ethynyl-2′-deoxyuridine | FUJIFILM Wako | Cat# 050-08844 |
| Click-iT™ Plus EdU Alexa Fluor™ 555 imaging kit | Invitrogen | Cat# C10638 |
| VECTASHIELD® antifade mounting medium | Vector Laboratories | Cat# H-1000 |
| **Software** | | |
| Prism (version 9.5.0) | GraphPad Software | https://www.graphpad.com/features |
| Fiji/ImageJ | Fiji/ImageJ | https://imagej.net/software/fiji/ |
| Adobe Photoshop | Adobe | https://creativecloud.adobe.com/ja/ |
| Adobe Illustrator | Adobe | https://creativecloud.adobe.com/ja/ |
| **Other** | | |
| Confocal laser microscope Zeiss LSM800 | Zeiss | LSM800 |
| Electric microinjector system | Narishige | IM-300 |
| Electroporator | BEX | CUY-21 |
| Electrodes | BEX | LF650P5 |

performed using the mouse reference genome (MM10), and Loupe Browser 6.4.0 was used for data analysis. *Kif23* expression was determined based on unique molecular identifier (UMI) counts in each spot, and violin plots were used to display *Kif23* expression in each cluster.

## Data processing of scRNA-seq analysis

All scRNA-seq data used in this study are available from the public databases: mouse cortex: GSE123335 (Loo et al, 2019), human cortex: phs001836 (Polioudakis et al, 2019; data was obtained from http://geschwindlab.dgsom.ucla.edu/pages/codexviewer), and ferret cortex: GSE234305 (Del-Valle-Anton et al, 2024; data was obtained from https://in.umh-csic.es/en/cortevo/). Normalization, dimensionality reduction, clustering, and visualization of scRNA-seq data were performed by Seurat (v4.3, Satija et al, 2015) function of NormalizeData, RunPCA, RunUMAP, FindNeighbors, FindCluster with default parameters.

## Animals

C57BL/6J mice (CLEA Japan) were used as wild-type (WT) in this study. Embryonic day 0.5 (E0.5) was defined as midday on the day when a vaginal plug was detected. All experiments were carried out in compliance with the National Institutes of Health guidelines for the care and use of laboratory animals, and experimental procedures were approved by the Ethics Committee for Animal Experiments of Tohoku University Graduate School of Medicine (2019 MdA-018-07).

## In situ hybridization

The dorsal telencephalon tissue obtained from WT E14.5 mice was used for RNA extraction using RNeasy Plus Mini Kit (QIAGEN). The extracted RNA was used for cDNA synthesis using the SuperScipt™III

First-Strand Synthesis System for RT-PCR (Invitrogen). cDNA fragment encoding *Kif23* (NM_024245, nucleotides 866-1367) was cloned into pBluescript II SK (–) (Stratagene). Digoxigenin-labeled RNA probe was synthesized using DIG RNA labeling kit (Roche). In situ hybridization on the frozen section was performed as previously described (Kikkawa et al, 2013). BZ-X710 fluorescence microscopy system (KEYENCE) was used to capture images.

## Immunohistochemistry

Immunohistochemistry was performed as described previously (Kikkawa et al, 2013). The frozen brain sections were incubated with primary antibodies diluted with 3% bovine serum albumin (BSA) in Tris-buffered saline with 0.1% Triton X-100 (TBS) with 0.1% Triton X-100 overnight at 4 °C. The following day, the sections were washed three times with TBS and then incubated with secondary antibodies and DAPI in the blocking solution. One hour later, sections were washed three additional times with TBS, mounted with VECTASHIELD® antifade mounting medium, and sealed with coverslips. 4′,6-Diamidino-2-phenylindole dihydrochloride (DAPI) (1:1000; Sigma) was used as nuclear counterstaining. Detailed information about the antibodies is provided in the reagents and tools table. All images were acquired using a confocal laser microscope Zeiss LSM800 (Carl Zeiss).

## In-utero electroporation

In-utero electroporation (IUE) was performed as described previously with slight modification (Nagai et al, 2022). Pregnant WT mice (E14.5) were anesthetized with isoflurane. A mixture of siRNA and *pCAG-EGFP* (final concentration of 2 µg/µl and 0.5 µg/µl, respectively) was injected into the lateral ventricle of the brain by using a glass capillary with the electric microinjector system (IM-300, Narishige). Five electric pulses at 40 V with a duration of 50 ms per pulse at 1 s intervals were applied

through the uterus using forceps-type electrodes (LF650P5, BEX) connected to an electroporator (CUY-21, BEX).

## EdU labeling

EdU labeling was performed as described previously with slight modification (Ji et al, 2017). 5-Ethynyl-2'-deoxyuridine (EdU) (50 mg/kg) was injected intraperitoneally into pregnant mice at the gestational day 15 or 16. For cell proliferation analysis, pregnant mice were sacrificed 2 h after the injection at the gestational day 16. For cell cycle exit analysis, the pregnant mice were sacrificed 24 h after the injection. EdU immunostaining was performed according to the Click-iT™ Plus EdU Alexa Fluor™ 555 imaging kit protocol (Invitrogen).

## Whole mount immunohistochemistry

Whole mount immunohistochemistry was performed as described previously with minor modifications (Shinohara et al, 2013). Electroporated embryos at E15.5 were perfused with 4% paraformaldehyde, and their brains were isolated and fixed in the same fixative for 25 min. Dorsolateral parts of the cerebral cortex were dissected, fixed for 2 min, and incubated in a blocking solution (3% BSA) for 1 h at room temperature. The samples were then incubated overnight at 4 °C with primary antibodies, including chicken anti-GFP (1:1000; Abcam) and rabbit anti-ZO-1 (1:1000; Invitrogen). After incubation with secondary antibodies, whole mounts were cover-slipped with VECTASHIELD® antifade mounting medium (Vector Laboratories).

## Spindle orientation measurement

Spindle orientation measurement was done following the protocol described previously with minor modifications (Yingling et al, 2008). Briefly, the angle θ of the mitotic cleavage plane of cells in metaphase or anaphase was measured by tracing a line along the GFP layer morphology that bisects the nuclei relative to the surface plane of the ventricular zone. The cells in metaphase or anaphase were discerned based on the nuclear morphology stained with DAPI. The angle α of spindle orientation was calculated as 90 degrees minus the angle θ. Analysis was performed by using the measurement tools of the Fiji software (National Institute of Health).

## Construction of human *KIF23* plasmids

Human *KIF23* cDNA was synthesized based on NCBI sequencing data (accession no. NM_001367805) (eurofins). The ORF of human *KIF23* was cloned within *pCAX* vector using In-Fusion system (TaKaRa). The disease associated c.755 T > A mutation was introduced to *human KIF23* cDNA by the PCR-based mutagenesis as described (Xia et al, 2015). Instead of Phusion DNA polymerase, KOD plus DNA polymerase (TOYOBO) was used. The mutation was confirmed by the Sanger sequencing.

## Statistical analysis

The cell counts were carried out using Fiji software (National Institute of Health). The fluorescence intensity of the spindle areas, midzone, and apical side was also measured using the same software. To measure α-tubulin intensity, the fluorescence intensity of α-tubulin in spindle areas was only assessed in the GFP⁺ metaphase cells with approximately similar γ-tubulin signal in the spindle poles. For ECT2 intensity measurements in the spindle midzone, the midzone area was defined based on the morphology of the GFP layer in the GFP⁺ anaphase cells. Regarding RhoA intensity measurements, the intensity of RhoA within 30 µm from the GFP⁺ ventricular surface was considered as the apical RhoA intensity. These mean fluorescence intensity values of the region of interest were normalized by subtracting the background mean fluorescence intensity. Statistical analyses were conducted using GraphPad Prism (version 9.5.0), including Student's $t$ tests, multiple unpaired $t$-tests, one-way ANOVA with Bonferroni's multiple comparisons test, and two-way ANOVA with Bonferroni's multiple comparisons test. Differences with $p < 0.05$ were considered statistically significant. $P$ values are denoted as follows: $*p < 0.05$, $**p < 0.01$, $***p < 0.001$, $****p < 0.0001$.

## Data availability

This study includes no data deposited in external repositories.

The source data of this paper are collected in the following database record: biostudies:S-SCDT-10_1038-S44318-024-00327-7.

## Peer review information

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

## Acknowledgements

We would like to thank Nobutaka Hirokawa, Yoshio Wakamatsu, Hiroshi Shinohara, and Shohei Ochi for their technical advice and valuable comments. We are grateful to Ms. Sayaka Makino for animal care and technical support. We thank all other members of the Osumi laboratory for their valuable comments and discussion. This work was supported by JSPS KAKENHI funding (#18K14999, 23K06297) to TK and AMED (#JP21wm0425003) to NO. A part of this study was supported by the Support System for Young Researchers to use research equipment, instruments, and devices at Tohoku University Core Facility Center.

## Author contributions

**Sharmin Naher**: Conceptualization; Data curation; Formal analysis; Validation; Investigation; Visualization; Methodology; Writing—original draft; Writing—review and editing. **Kenji Iemura**: Validation; Investigation; Methodology; Writing—review and editing. **Satoshi Miyashita**: Formal analysis; Investigation; Methodology; Writing—review and editing. **Mikio Hoshino**: Investigation; Methodology; Writing—review and editing. **Kozo Tanaka**: Validation; Methodology; Writing—review and editing. **Shinsuke Niwa**: Investigation; Methodology; Writing—review and editing. **Jin-Wu Tsai**: Supervision; Investigation; Methodology; Writing—original draft; Writing—review and editing. **Takako Kikkawa**: Conceptualization; Supervision; Funding acquisition; Validation; Investigation; Visualization; Methodology; Writing—original draft; Writing—review and editing. **Noriko Osumi**: Conceptualization; Supervision; Funding acquisition; Validation; Investigation; Methodology; Writing—original draft; Writing—review and editing.

Source data underlying figure panels in this paper may have individual authorship assigned. Where available, figure panel/source data authorship is listed in the following database record: biostudies:S-SCDT-10_1038-S44318-024-00327-7.

## Disclosure and competing interests statement

The authors declare no competing interests.

# Expanded View Figures

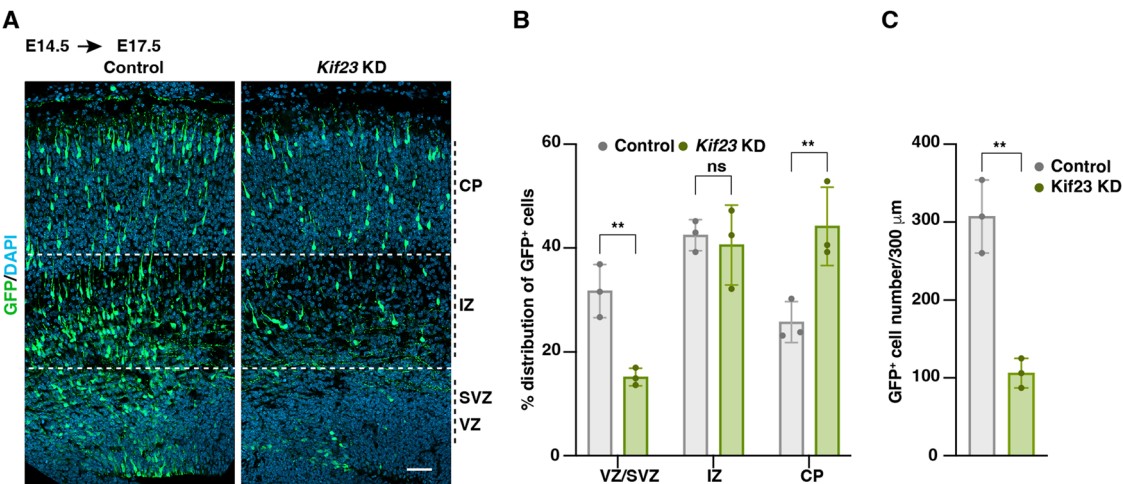

**Figure EV1.  Knockdown of *Kif23* leads to profound loss of GFP expressing cells.**

(A) Representative images of GFP$^+$ cells distribution in the control and *Kif23*-KD cortices at E17.5. Dashed lines illustrate the borders among VZ/SVZ, IZ, and CP. Scale bar, 50 μm. (B) Quantification of GFP$^+$ cells distribution in VZ/SVZ, IZ, and CP, respectively within the 300 μm wide column of E17.5 cortices. The data represent the mean ± SD Control *n* = 921 cells, 3 embryos; *Kif23*-KD *n* = 318 cells, 3 embryos). Two-way ANOVA with Bonferroni's multiple comparison test, *p* values from left to right: **$p$ = 0.007258, **$p$ = 0.003357, ns: not significant. (C) Quantification of the average number of GFP$^+$ cells within the 300 μm wide column of E17.5 cortices. The data represent the mean ± SD (Control *n* = 921 cells, 3 embryos; *Kif23*-KD n = 318 cells, 3 embryos). Two-tailed Student's *t* test, **$p$ = 0.002355.

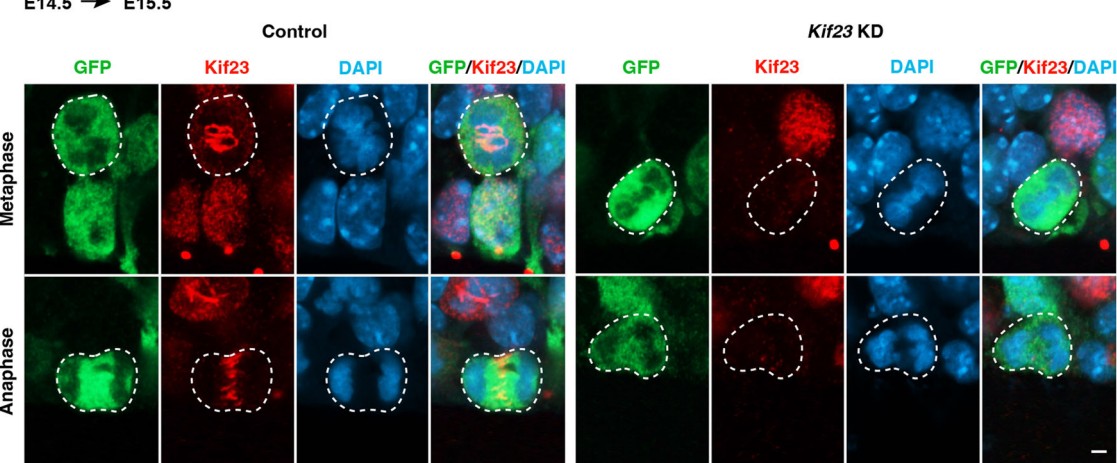

**Figure EV2.  *Kif23* knockdown reduces Kif23 protein level in the microtubule.**

Representative images of the control and *Kif23*-KD cortical sections at E15.5 stained for GFP, Kif23, and DAPI. Examples of GFP$^+$ cells at the metaphase or anaphase stage are outlined. Scale bar, 2 μm.

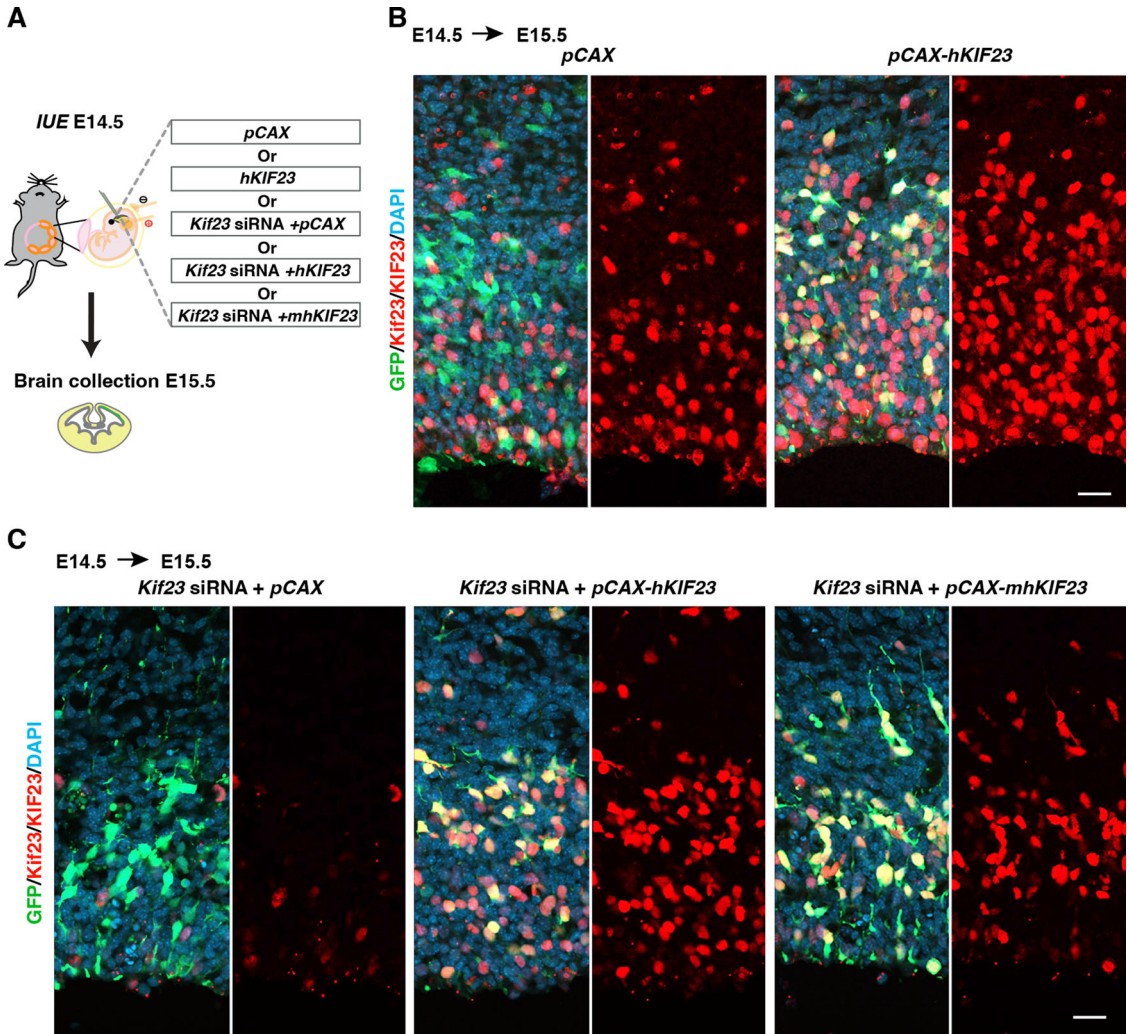

**Figure EV3. *Kif23* siRNA decreases mouse Kif23 protein levels, not human KIF23.**

(**A**) Schematic overview of the experiment. (**B**) Representative images of mouse cortices one day after electroporation of *pCAX* or *pCAX-hKIF23* stained for GFP and Kif23/KIF23. Scale bar, 20 μm. (**C**) Representative images of mouse cortices one day after electroporation of *Kif23* siRNA/*pCAX*, *Kif23* siRNA/*pCAX-hKIF23* or *Kif23* siRNA/*pCAX-mhKIF23* stained for GFP and Kif23/KIF23. Scale bar, 20 μm.

