## [Peer Review File · The EMBO Journal]

Kinesin-like motor protein KIF23 maintains the neural and progenitor cell pool in the developing neocortex

Sharmin Naher, Kenji Iemura, Satoshi Miyashita, Mikio Hoshino, Kozo Tanaka, Shinsuke Niwa, Jin-Wu Tsai, Takako Kikkawa, and Noriko Osumi

Corresponding author(s): *Noriko Osumi (osumi@med.tohoku.ac.jp)* , *Takako Kikkawa (kikkawa@med.tohoku.ac.jp)*

Review Timeline:

Submission Date:	10th Apr 24
Editorial Decision:	3rd Jun 24
Revision Received:	30th Sep 24
Editorial Decision:	4th Nov 24
Revision Received:	15th Nov 24
Accepted:	15th Nov 24

Editor: Ioannis Papaioannou

Transaction Report:

Dear Prof. Osumi,

Thank you for submitting your manuscript EMBOJ-2024-117529 for consideration by The EMBO Journal, and for your patience during peer review. It has now been seen by two experts in the field, and we have received their comprehensive reports, which you can find below.

As you will see, both referees are very positive about the study and recognize the novelty and significance of the findings, the proper design and quality of the work, and the presentation of the data. They are both supportive of publication of this study in The EMBO Journal provided that a number of constructive concerns they raise and useful suggestions they make for strengthening the study and the manuscript further will be successfully addressed in a thorough revision.

Given the referees' positive comments and recommendations, I would like to invite you to submit a revised version of the manuscript along with a detailed point-by-point response addressing all referees' comments. I should add that it is EMBO Journal policy to allow only a single round of major revision, and acceptance of your manuscript will therefore depend on the completeness of your responses in this revised version. Please let me know if you have any questions or comments that you would like to discuss with me.

We generally allow three months as standard revision time (September 2, 2024). As a matter of policy, competing manuscripts published during this period will not negatively impact our assessment of the conceptual advance presented by your study. However, we request that you contact us as soon as possible upon publication of any related work, to discuss how to proceed. Should you foresee a problem in meeting this three-month deadline, please let us know in advance and we may be able to grant an extension.

Thank you for the opportunity to consider your work for publication in The EMBO Journal. I look forward to your revision.

Best regards,

Ioannis

Instructions for preparing your revised manuscript

1. When you are ready to submit the revision, please upload:

- A Word file of the manuscript text (including legends of main Figures, EV Figures and Tables). Please make sure that changes are highlighted (or "tracked") to be clearly visible.

- Individual production-quality figure files (one file per figure). When assembling your figures, please refer to our figure preparation guidelines in order to ensure proper formatting and readability in print as well as on screen:

If the data shown in a figure are obtained from n {less than or equal to} 2, please use scatter plots showing the individual data points.

- i. the name of the statistical test used to generate error bars and P values
- ii. the number (n) of independent experiments (please specify technical or biological replicates) underlying each data point (discussion of statistical methodology can be reported in the Materials and Methods section, but figure legends should contain a basic description of n , P , and the test applied)
- iii. the nature of the bars and error bars (s.d., s.e.m.).

- A point-by-point response to the referees' comments, with a detailed description of the changes made (as a word file). All referees' concerns must be fully addressed and their suggestions taken on board. When preparing your letter of response to the referees' comments, please bear in mind that this will form part of the Review Process File and will therefore be available online

to the community. Please note that you have the possibility to opt out of the transparent process at any stage prior to publication by letting the editorial office know (contact@embojournal.org); if you do opt out, the Review Process File link will point to the following statement: "No Review Process File is available with this article, as the authors have chosen not to make the review process public in this case.". For more details on our Transparent Editorial Process, please visit our website: <https://www.embopress.org/page/journal/14602075/authorguide#transparentprocess>

- Expanded View (EV) files (replacing Supplementary Information) that are collapsible/expandable online. A maximum of 5 EV Figures can be typeset. EV Figures should be cited as "Figure EV1, Figure EV2" etc. in the text, and their respective legends should be included in the manuscript file after the legends of regular figures. See detailed instructions regarding Expanded View files here:

- For the figures that you do NOT wish to display as Expanded View figures, they should be bundled together with their legends in a single PDF file called "Appendix", which should start with a short Table of Contents (including page numbers). Appendix figures should be referred to in the main text as: "Appendix Figure S1, Appendix Figure S2" etc. Please see detailed instructions here: <https://www.embopress.org/page/journal/14602075/authorguide#expandedview>

- A complete author checklist, which you can download from our author guidelines (<https://www.embopress.org/page/journal/14602075/authorguide>). Please note that the checklist will also be part of the Review Process File.

2. Please note that no statistics should be calculated and shown in Figures if $n=2$. Please also note that each p value should be reported as an exact value.

3. Before submitting your revision, primary datasets (and computer code, where appropriate) produced in this study need to be deposited in appropriate public databases (see <https://www.embopress.org/page/journal/14602075/authorguide#dataavailability>).

*** The Data Availability Section is restricted to new primary data that are part of this study. In case you have no data that require deposition in a public database, please state so instead of referring to the database: "Our study includes no data deposited in public repositories." under the heading "Data availability". ***

If you have new data that require deposition in a public database, the accession numbers and databases should be listed in a formal "Data availability" section (placed after Materials and Methods) that follows the model below (see also <https://www.embopress.org/page/journal/14602075/authorguide#dataavailability>):

Data availability

- RNA-seq data: Gene Expression Omnibus GSE46843 (<https://www.ncbi.nlm.nih.gov/geo/query/acc.cgi?acc=GSE46843>)
- [data type]: [name of the resource] [accession number/identifier/doi] ([URL or identifiers.org/DATABASE:ACCESSION])

*** All links should resolve to a page where the data can be accessed. ***

*** Please remember to provide in the Data availability section of your revised manuscript reviewer passwords if the datasets are not yet public. ***

4. Please check that the title and the abstract of the manuscript are brief, yet explicit, even to non-specialists. The length of the title should not exceed 100 characters, and the abstract should be a single paragraph not exceeding 175 words.

5. Please also note our reference format: <https://www.embopress.org/page/journal/14602075/authorguide#referencesformat>.

7. Please remember: digital image enhancement is acceptable practice, as long as it accurately represents the original data and conforms to community standards. If a figure has been subjected to significant electronic manipulation, this must be noted in the figure legend or in the "Materials and Methods" section. The editors reserve the right to request original versions of figures and the original images that were used to assemble the figure.

8. Our journal encourages inclusion of data citations in the reference list to directly cite datasets that were obtained from public databases. Data citations in the article text are distinct from normal bibliographical citations and should directly link to the database records from which the data can be accessed. In the main text, data citations are formatted as follows: "Data ref:

Smith et al, 2001" or "Data ref: NCBI Sequence Read Archive PRJNA342805, 2017". In the Reference list, data citations must be labeled with "[DATASET]". A data reference must provide the database name, accession number/identifiers, and a resolvable link to the landing page from which the data can be accessed at the end of the reference. Further instructions are available at: <https://www.embopress.org/page/journal/14602075/authorguide#referencesformat>.

9. We request authors to consider both actual and perceived competing interests. Please review our policy (<https://www.embopress.org/page/journal/14602075/authorguide#conflictsofinterest>) and update your competing interests statement if necessary. Please name this section 'Disclosure and competing interests statement' and place it after the Acknowledgements section.

10. Please note that all corresponding authors are required to provide an ORCID ID upon submission of a revised manuscript (<https://orcid.org/>). Please find instructions on how to link your ORCID ID to your account in our manuscript tracking system in our Author guidelines (<https://www.embopress.org/page/journal/14602075/authorguide#authorshipguidelines>).

11. We use CRediT to specify the contributions of each author in the journal submission system. CRediT replaces the author contribution section, which should be removed from the manuscript. Please use the free text box to provide more detailed descriptions. See also guide to authors: <https://www.embopress.org/page/journal/14602075/authorguide#authorshipguidelines>.

13. We would also welcome the submission of cover suggestions or motifs to be used by our Graphics Illustrator in designing a cover.

14. Please use the link below to submit your revision:
<https://emboj.msubmit.net/cgi-bin/main.plex>

Referee #1:

This is a very nice paper studying the role on cerebral cortex development of the centrosome-related protein Kif23, mutated in human microcephaly. The authors describe that Kif23 is expressed in cortical progenitor cells during the developmental period of neurogenesis, and then investigate its role by means of acute loss-of-function experiments using RNA interference and in utero electroporation. Loss of Kif23 leads to disrupted mitotic spindle orientation and impaired cytokinesis at the end of mitosis, linked to precocious cell cycle exit with apoptosis of newborn neurons. The authors further describe that loss of Kif23 disrupts the localization of apical junction proteins and thus the integrity of the apical surface of cortical NPCs. Last, they show that some of these phenotypes are rescued by human KIF23 but not its mutated form linked to microcephaly, suggesting that these defects underlie the occurrence of human microcephaly. The study is very well designed, executed and presented, with beautiful images and quantification of the key phenotypes. I only have some points that should be clarified, and few suggestions for further improvement of an already excellent work.

1- The analysis of Kif23 expression based on scRNA-seq is very nice, but it remains unresolved whether Kif23 is expressed only in RGCs or also in IPCs. Top2a marks mitotic cells, but these include both types of NSC: RGC and IPC. Immunostains and ISH do not clarify this point, because the authors did not quantify %Ki23+ cells that are Pax6+ vs Tbr2+. The fact that the authors find a lower proportion of GFP+Tbr2+ cells upon Kif23-KD is consistent with this notion.

2- If EdU and PH3 analyses were quantified over total GFP (Figure 3), this is quite non-informative given the very strong difference in distribution of GFP+ cells across layers. In such case, the analysis must instead be done over VZ cells.

3- The analysis presented in Fig 2 E-J should be done at E15, for consistency with analyses in Fig. 2A-D and because the phenotype of cell misplacement is already very dramatic at E15, so one must investigate the primary effects, not effects potentially secondary already by E16.

4- Regarding the analyses of apoptosis, the co-localization stains of CC3 cells with Tuj1 or Hu is very unconvincing. Perhaps the authors can test nuclear neuronal markers, as for progenitors, for example NeuroD2.

5- Page 10: "These micronuclei and pyknotic doublets closely resemble binucleated cells, a phenomenon known to result from improper cytokinetic events. Consequently, our results put forth the intriguing notion that a subset of cells fails to complete

cytokinesis in Kif23-deficient NSPCs, shedding light on the intricacies of Kif23's role in cell division and survival." - This is very intriguing indeed, especially considering the increase in Tuj1+ cells in VZ alongside with the decrease in Sox2+ cells. Are these doublets of failed cytokineses the excess cells in VZ that are Tuj1+ and also CC3? If that is the case, is this defect affecting mitoses from RGCs and also IPCs?

6- Related to the previous point, on page 11 the authors state: "This dual impact of Kif23 deficiency on both cell survival and cell cycle progression underscores its multifaceted role in regulating NSPC behavior in the developing cortex." - This is confusing, as the previous section highlights that the phenotype is a defect on newborn neurons entering apoptosis, but now we are told that it is a defect of cell-cycle progression on NSCs. This needs to be better explained.

7- The size of apical domain of individual cells in the cerebral cortex increases significantly in Kif23-KD. Does this mean loss of apical junction integrity, or maybe increased number of fused apical cells, more consistent with the abundance of doublets? Apical instability frequently leads to apical detachment and delamination; is this observed in Kif23-KD?

8- The authors explore a publish dataset from human embryonic cortex to compare with mouse and their own results here. A recent scRNA-seq study in ferret identifies a large number of RGC and IP subtypes (Science Advances), using in part also this dataset from Polioudakis. It would be interesting to check in that more detailed analysis if Kif23 is also expressed in only RGCs in S/G2/M.

9- Are differences between si-Kif23 and si-Kif23+mhKIF23 significant in Fig 7D? The authors should provide some plausible explanation for this partial rescue  mutant version is partly functional...

10- It is unclear whether the reduction in Tbr2+ cells is a failure to generate them from RGCs, or the result of their apoptosis. This should be better clarified. Also, it is not clear if the nuclear doublets were observed only in neurons, or only in NSCs, or in both.

Minor points:

Images in Fig 2C,D should show the VZ surface, for a better assessment of its integrity overall. The difference observed is very dramatic. The authors should check for apoptosis following si-Kif23. In fact, the authors do perform such very necessary analysis in Figure 5, in a very nice manner, but it should be presented before, where it fits much better with the observations and line of argumentation presented in Figure 2.

Figure 6C,D, does it show activated p53? If yes, must indicate.

The analysis of DNA damage, p53 and p21 expression must be done in all cases in VZ/SVZ and IZ, for consistency.

Referee #2:

In this manuscript, the authors aim to elucidate how Kif23, a member of the kinesin family, contributes to the formation of mammalian cerebral structures. They focused on the role of Kif23 in the developing neocortex of mouse fetal brains. The authors found that Kif23 is highly expressed in neural stem/progenitor cells of the neocortical ventricular zone during a specific period from E12.5 to E15.5, when neurogenesis is active in the mouse fetal brain. At E14.5, they performed in utero electroporation of siRNA to suppress the expression of Kif23. Through histological analysis, the authors found that the loss of Kif23 expression leads to progenitor cells differentiating into neuronal cells earlier and migrating to the Intermediate Zone. They also showed that Kif23 knockdown results in several abnormalities, including disorganized spindles, changes in the direction of cell division, an increase in apoptotic cells, and premature exit from the cell cycle. These abnormalities contribute to early neuronal differentiation, cell migration, and a subsequent reduction in the number of neurons. Additionally, they demonstrated that knockdown of Kif23 results in abnormalities in the apical cell adhesion in the ventricular zone (VZ) of the E15.5 embryonic brain tissue. Finally, and most interestingly, they showed that the phenotype induced by Kif23 knockdown, where progenitor cells prematurely become neuronal cells and migrate to the Intermediate Zone, can be rescued by the expression of human Kif23. However, this rescue is not achieved with the mutated Kif23, which is known to cause microcephaly in humans.

Overall, the data suggest that Kif23 plays a crucial role in properly controlling the division orientation and cytokinesis of neural progenitor cells, thereby maintaining the proliferative capacity of progenitor cells. It is particularly interesting that the study provides insight into the mechanism by which mutations in Kif23, found in humans, can cause microcephaly. Most of the authors' conclusions are supported by sufficient quality and quantitative data, making the manuscript suitable for publication in EMBO J. However, some interpretations of the data are based on unclear evidence, and there are deficiencies in the presentation of statistical data. Therefore, I recommend that the following points be addressed or further explained before the manuscript is accepted for publication.

Major points

- 1) The authors claim that Kif23KD induces spindle disorganization. However, there is no specific description of the abnormalities and no clear criteria for determining "disorganized spindle." From the provided images (FigC4, EV1), it is unclear what abnormalities are present. For instance, the spindle morphology in Fig4C appears slightly different from the control, but it might simply be due to the spindle being angled relative to the focal plane of the microscope. Given that the authors demonstrate changes in spindle angle due to Kif23KD in Fig4E-H, careful examination is needed to distinguish whether the observed spindle disorganization is due to viewing angle or actual morphological changes. Comparing spindles where gamma tubulin signals are visible at both poles could clarify this issue. Additionally, if significant spindle abnormalities occur in prometaphase/metaphase, the spindle assembly checkpoint should reduce the proportion of cells progressing to anaphase. If the authors strongly assert the occurrence of disorganized spindles, supporting data should be provided.
- 2) In Fig5I, the RhoA signal appears to be reduced regardless of GFP positivity. To substantiate the claim that Kif23KD leads to decreased RhoA expression, quantitative data is necessary. Furthermore, for the series of experiments shown in Fig5H, J, K, quantitative data is also required, as these experiments form a crucial part of the basis for the authors' conclusions.
- 3) The manuscript currently lacks information on the sample size for each experimental condition, and instead, the number of experimental repeats appears to be incorrectly denoted as "n=X". Typically, the number of repeats should be denoted as "N". Please include the sample size (n) for each group or condition. This information is essential for evaluating the statistical power and reproducibility of the results.

Minor points

- The figure citations on page 7 appear to be incorrect. Where it says (Figs. 2C, D), it should be (Figs. 2C, E), and where it says (Figs. 2E, F), it should be (Figs. 2D, F).
- On page 7, the text describing Fig2 states "at E15.5," but the figure and legend indicate "E16.5." Please correct this inconsistency.

Point-by-Point Response

Referee #1:

This is a very nice paper studying the role on cerebral cortex development of the centrosome-related protein Kif23, mutated in human microcephaly. The authors describe that Kif23 is expressed in cortical progenitor cells during the developmental period of neurogenesis, and then investigate its role by means of acute loss-of-function experiments using RNA interference and in utero electroporation. Loss of Kif23 leads to disrupted mitotic spindle orientation and impaired cytokinesis at the end of mitosis, linked to precocious cell cycle exit with apoptosis of newborn neurons. The authors further describe that loss of Kif23 disrupts the localization of apical junction proteins and thus the integrity of the apical surface of cortical NPCs. Last, they show that some of these phenotypes are rescued by human KIF23 but not its mutated form linked to microcephaly, suggesting that these defects underlie the occurrence of human microcephaly. The study is very well designed, executed and presented, with beautiful images and quantification of the key phenotypes. I only have some points that should be clarified, and few suggestions for further improvement of an already excellent work.

We appreciate the reviewer's positive assessment of our work and constructive comments. We have incorporated new experimental data and quantification to address specific concerns.

Major points:

1. The analysis of Kif23 expression based on scRNA-seq is very nice, but it remains unresolved whether Kif23 is expressed only in RGCs or also in IPs. Top2a marks mitotic cells, but these include both types of NSC: RGC and IP. Immunostains and ISH do not clarify this point, because the authors did not quantify %Kif23+ cells that are Pax6+ vs Tbr2+. The fact that the authors find a lower proportion of GFP+Tbr2+ cells upon Kif23-KD is consistent with this notion.

Response: In response to the request by Reviewer 1, we have now quantified the proportion of Kif23⁺Sox2⁺ and Kif23⁺Tbr2⁺ cells in the mouse cortex. The new data, i.e. double immunostaining for Kif23/Sox2 and Kif23/Tbr2, indicate that the majority of Kif23⁺ cells were Sox2⁺ (89.4%±2.2%, n=815 cells, 4 embryos), while Kif23 expression was also detected in Tbr2⁺ cells (19.9%±1.7%, n=784 cells, 4 embryos) (**Appendix Fig. S1A, B**). Although the proportion of Kif23⁺Tbr2⁺ cells was less than that of Kif23⁺Sox2⁺ cells, this result suggests the possibility that a lower proportion of GFP⁺Tbr2⁺ cells due to Kif23-KD could result from the effects of Kif23 on a part of the IPs as well as RGCs. We have added these new figures (**Appendix Fig. S1A, B**) and the description in the Results and Discussion in the revised manuscript (p. 6, lines 126-129, p. 15-16, lines 407-414).

2. If EdU and PH3 analyses were quantified over total GFP (Figure 3), this is quite non-informative given the very strong difference in distribution of GFP⁺ cells across layers. In such case, the analysis must instead be done over VZ cells.

Response: We agree with the reviewer's comment and have conducted quantitative analyses of EdU⁺GFP⁺ and GFP⁺PH3⁺ cells relative to the GFP⁺ cells only in the VZ (**Fig. 4F, H**). The new quantitative data clearly showed a significant reduction in EdU⁺GFP⁺ and GFP⁺PH3⁺ cells in the VZ of E16.5 cortices after Kif23 KD compared to the control group, indicating that Kif23 KD

impairs cell proliferation. We have modified these data in **Fig. 4F, H** and corresponding description in the Results of the manuscript (p. 9, lines 202-208).

3. The analysis presented in Fig 2 E-J should be done at E15, for consistency with analyses in Fig. 2A-D and because the phenotype of cell misplacement is already very dramatic at E15, so one must investigate the primary effects, not effects potentially secondary already by E16.

Response: We appreciate the reviewer's suggestion. We investigated cell proliferation and mitotic activity one day after *Kif23*-KD, at E15.5. As a result, we consistently observed a significant reduction in cell proliferative capacity and mitotic activity at E15.5 as well as at E16.5 (**Appendix Fig. S3**). These results suggest that the proliferation and mitotic activity were affected by *Kif23*-KD as the primary effect. We have added a new figure (**Appendix Fig. S3**) and the corresponding description in the Results (p. 9, lines 202-208).

4. Regarding the analyses of apoptosis, the co-localization stains of CC3 cells with Tuj1 or Hu is very unconvincing. Perhaps the authors can test nuclear neuronal markers, as for progenitors, for example NeuroD2.

Response: We acknowledge the reviewer's legitimate criticism. We have thus performed co-staining of CC3 with a nuclear neuronal marker NeuroD2 in brain slices. Although we observed *Tuj1*⁺*CC3*⁺ premature neuronal apoptosis, pyknotic cells were rarely positive for NeuroD2 (12.3%±3.4%, n=200 cells, 4 embryos) (**Appendix Fig. S2C**). Considering the expression of NeuroD2 in relatively late differentiating neurons than that of *Tuj1* (Sagner et al., PLoS Biol, 2021) and its transient expression (Farah et al., Development, 2000), it is possible that the differentiated cells undergo apoptosis and are removed before expressing NeuroD2. We have newly added a figure (**Appendix Fig. S2C**) and the corresponding detailed description in the Results (p. 8, lines 182-186).

5. Page 10: "These micronuclei and pyknotic doublets closely resemble binucleated cells, a phenomenon known to result from improper cytokinetic events. Consequently, our results put forth the intriguing notion that a subset of cells fails to complete cytokinesis in Kif23-deficient NSPCs, shedding light on the intricacies of Kif23's role in cell division and survival." - This is very intriguing indeed, especially considering the increase in Tuj1+ cells in VZ alongside with the decrease in Sox2+ cells. Are these doublets of failed cytokineses the excess cells in VZ that are Tuj1+ and also CC3? If that is the case, is this defect affecting mitoses from RGCs and also IPCs?

Response: Related with the response #4, we found by co-staining of *Tuj1* and *CC3* that the majority of pyknotic doublets in the VZ were double positive for *Tuj1* and *CC3* (76.5%±3.7%, n = 103 cells, 4 embryos) (**Appendix Fig. S2A, B**). Furthermore, we noted a significant reduction in apical mitotic cells in the VZ, while the number of abapical mitotic cells in the VZ/SVZ remained unchanged (**Appendix Fig. S3C, E**). Since *Kif23* is predominantly expressed in mitotic RGCs and, to a lesser extent, in IPCs (**Appendix Fig. S1A, B**), it is possible that this defect mainly affects mitosis in RGCs and, to some extent, in IPCs. Still, it is difficult to completely exclude the possibility of affection to IPC by *Kif23*-KD because the IPCs also

express Kif23 (**Appendix Fig. S1A, B**). Therefore, we have addressed this point in the Discussion in the revised manuscript (p. 15-16, lines 407-414).

6. Related to the previous point, on page 11 the authors state: "This dual impact of Kif23 deficiency on both cell survival and cell cycle progression underscores its multifaceted role in regulating NSPC behavior in the developing cortex." - This is confusing, as the previous section highlights that the phenotype is a defect on newborn neurons entering apoptosis, but now we are told that it is a defect of cell-cycle progression on NSCs. This needs to be better explained.

Response: We understand the reviewer's confusion about our statement. Various studies using cell lines have demonstrated that cultured cells experiencing cytokinetic failure after exiting mitosis can progress into the interphase state, express p53, and subsequently activate the p53-dependent re-replication checkpoint to inhibit DNA replication reinitiation (Minn et al., *Genes Dev*, 1996; Lanni & Jacks, *Mol Cell Biol*, 1998; Hayashi & Karlseder, *Oncogene*, 2013; S Pedersen et al., *Nat Commun*, 2016). In accordance with these previous studies and our own findings, we proposed the following: After knockdown of *Kif23*, NSPCs were unable to complete cytokinetic division. Consequently, cells with failed cytokinesis, upon attempting to re-enter the cell cycle, activated the γ -H2AX-p53-p21 signaling pathway in the subsequent cell cycle. This pathway prevented the G1-S phase transition, prompting cells to exit the cell cycle, differentiate into neurons, and ultimately undergo apoptosis. We have revised a part of the Results to make these points clearer (p. 11, lines 281-286).

7. The size of apical domain of individual cells in the cerebral cortex increases significantly in Kif23-KD. Does this mean loss of apical junction integrity, or maybe increased number of fused apical cells, more consistent with the abundance of doublets? Apical instability frequently leads to apical detachment and delamination; is this observed in Kif23-KD?

Response: We appreciate the reviewer's constructive comment. To investigate whether the increase in apical domain size is due to a defect in apical integrity or the presence of doublets on the apical surface, we conducted a new analysis of nuclear morphology using DAPI staining on whole mounts of *Kif23*-KD cortices at E15.5. Our observations did not reveal the presence of doublets on the apical surface (**Appendix Fig. S4A**), which is consistent with the presence of doublet cells in the basal side of the VZ (not on the apical surface) and SVZ (**Fig. 3A**), indicating a potential compromise in junctional integrity at the apical surface.

The centrosome in NSPCs, a crucial component of the apical membrane, is anchored by interacting with microtubules and junctions of the apical domain (Yang et al., *Curr Opin Neurobiol*, 2021). Along with the disruption of apical integrity, the decrease in apical centrosomes and increase in the abventricular centrosomes are often considered as an indication of NSPCs delamination (Tavano et al., *Neuron*, 2018; Uzquiano et al., *Cell Rep*, 2019; Lin et al., *J Cell Sci*, 2020). Therefore, we have performed immunostaining with centrosomal protein γ -tubulin to quantify the number of apical and abventricular centrosomes (Shinohara et al., *Biol Open*, 2013). Following *Kif23*-KD, we found a significant reduction in the number of apical centrosomes but not in abventricular centrosomes in the VZ (**Appendix Fig. S4B, C**). One possible explanation for this result is that the observed increase in the apical domain size and the decrease in the number of apical centrosomes in the apical endfeet may couple with the unchanged abventricular centrosome in the VZ, indicating that the delaminated NSPCs may have

already differentiated into neurons. Alternatively, the disruption of apical integrity after *Kif23*-KD may not lead to NSPCs delamination, but rather, the decrease in the number of apical centrosomes may result from a decline in the proliferating cell population after *Kif23*-KD. We have thus added new figures (**Appendix Fig. S4A-C**) and the detailed description in the Results and Discussion in the revised manuscript (p. 12, lines 303-312, p. 16-17, lines 443-450).

8. The authors explore a publish dataset from human embryonic cortex to compare with mouse and their own results here. A recent scRNA-seq study in ferret identifies a large number of RGC and IP subtypes (Science Advances), using in part also this dataset from Polioudakis. It would be interesting to check in that more detailed analysis if Kif23 is also expressed in only RGCs in S/G2/M.

Response: In response to the reviewer's thoughtful suggestion, we assessed the expression of *Kif23* using the scRNA-seq datasets from the germinal zone of developing ferret cortices at embryonic day 34 and postnatal day 1 to determine its abundance in distinct cell types in the S/G2/M phases (Del-Valle-Anton et al., Sci Adv, 2024). We found that *Kif23* is specifically expressed in RGCs as well as in IPs clusters in the S/G2/M phases in the ferret (**Appendix Fig. S5**). This result correlates with our mouse immunostaining data (**Fig.1D, Appendix Fig. S1A, B**). We have added a new figure (**Appendix Fig. S5**) and described the Results and Discussion (p. 12-13, lines 320-327; p. 16, lines 414-420). A new Abstract now includes this issue (p. 3, line 53).

9. Are differences between si-Kif23 and si-Kif23+mhKIF23 significant in Fig 7D? The authors should provide some plausible explanation for this partial rescue  mutant version is partly functional...

Response: We understand the reviewer's critical point, and the population of GFP⁺ cells between si-*Kif23* and si-*Kif23*+*mhKIF23* was statistically different (**Fig 9D**; we have newly added error bars expressing all the statistical significances), suggesting that *mhKIF23* may retain the partial activity. To examine the possibility of mislocalization of *mhKIF23* in the dividing cells, we tried to detect the tagged proteins in a neuroblastoma cell line (NB2A) transfecting the construct of *hKIF23*- or *mhKIF23*-fused to the tag sequencing. However, it was difficult for us to detect the tagged protein in the cells transiently transfected with *hKIF23* or *mhKIF23*. It seems unfortunately that our experiment was not successful using this strategy, and it will be difficult during the revision period to try to newly isolate permanently proliferating cells that have the ability to express the *hKIF23* or *mhKIF23*.

Given that this mutation is a missense mutation and that severe microcephaly has only been identified in individuals homozygous for this mutation (Karaca et al., Neuron, 2015), it is possible that this missense mutation may partially suppress KIF23 function rather than completely eliminate it. It is also unclear whether this mutation in the motor domain of *hKIF23* affects motor activity. Therefore, we have carefully described the Results and the issues and hypotheses in the Discussion (p. 13, lines 346-349; p. 17, lines 460-472).

10. It is unclear whether the reduction in Tbr2+ cells is a failure to generate them from RGCs, or the result of their apoptosis. This should be better clarified. Also, it is not clear if the nuclear doublets were observed only in neurons, or only in NSCs, or in both.

Response: We acknowledge the reviewer's constructive comments. Based on our findings, we have pointed out the possible mechanism of the reduction in Tbr2⁺ cells after *Kif23*-KD.

About the possibility of failure to Tbr2⁺ cells generation from RGCs, we found that mitotic activity at E15.5 *Kif23*-KD cortices was significantly decreased in apical mitotic cells in the RGC-enriched apical surface, while the number of abapical mitotic cells remained unchanged in the IPs located in the abapical VZ and SVZ (**Appendix Fig. S3C, E**). This suggests that *Kif23*-KD mainly impairs the mitotic function of RGCs, but not of Tbr2⁺ IPs. The decrease in mitotic RGCs coupled with unchanged mitotic IPs indicates that the reduction of Tbr2⁺ cells in *Kif23*-KD cortices is likely due to a failure in their generation from RGCs.

With regard to the possibility of the apoptosis of Tbr2⁺ cells, we found predominant expression of *Kif23* in the RGCs (~90%) and to a lesser extent in IPs (~20%) (**Appendix Fig. S1A, B**) using our immunohistochemical analysis in the E14.5 mouse cortex. We also found that expression of *Kif23* in both RGCs and IPs in S/G2/M phases from our re-analysis of public scRNA-seq datasets from the developing ferret cortex (**Appendix Fig. S5**; Del-Valle-Anton et al., Sci Adv, 2024). Considering that pyknotic nuclei were detected not only in RGCs (~25%) but also in Tbr2⁺ IPs (~24%) (**Fig. 3D, E**), it is also possible that *Kif23*-KD mainly affects RGCs function. However, we cannot exclude the possibility that *Kif23*-KD may directly regulate apoptosis of a part of the IP population.

We have addressed above points in the Discussion in the revised manuscript (p. 15-16, lines 407-414).

Minor points:

Images in Fig 2C,D should show the VZ surface, for a better assessment of its integrity overall. The difference observed is very dramatic. The authors should check for apoptosis following si-Kif23. In fact, the authors do perform such very necessary analysis in Figure 5, in a very nice manner, but it should be presented before, where it fits much better with the observations and line of argumentation presented in Figure 2.

Response: In response to the reviewer's constructive suggestion, we have revised **Fig. 2C, D** by adding dashed lines to enhance the visibility of the VZ surface. Furthermore, we agree with the reviewer's comments and have rearranged figures by changing the order (i.e., repositioning **Fig. 5** as **Fig. 3**) and separating images of the apoptosis and cytokinetic molecules into **Fig. 3** and **Fig. 6**, respectively. We hope this revision meets the reviewer's expectations.

Figure 6C,D, does it show activated p53? If yes, must indicate.

Response: As the reviewer points out, we used the antibody for activated p53 in **Fig. 6C, 6D**. We indicate this information in the Results, Discussion, and Methods in the revised manuscript (p.11, 15, 20, lines 274, 402, 524).

The analysis of DNA damage, p53 and p21 expression must be done in all cases in VZ/SVZ and IZ, for consistency.

Response: We have analyzed the levels of γ -H2AX, p53, and p21 expression both in the VZ/SVZ and IZ regions, ensuring consistency across all cases (**Fig. 7B, D, F**).

Referee #2:

In this manuscript, the authors aim to elucidate how Kif23, a member of the kinesin family, contributes to the formation of mammalian cerebral structures. They focused on the role of Kif23 in the developing neocortex of mouse fetal brains. The authors found that Kif23 is highly expressed in neural stem/progenitor cells of the neocortical ventricular zone during a specific period from E12.5 to E15.5, when neurogenesis is active in the mouse fetal brain. At E14.5, they performed in utero electroporation of siRNA to suppress the expression of Kif23. Through histological analysis, the authors found that the loss of Kif23 expression leads to progenitor cells differentiating into neuronal cells earlier and migrating to the Intermediate Zone. They also showed that Kif23 knockdown results in several abnormalities, including disorganized spindles, changes in the direction of cell division, an increase in apoptotic cells, and premature exit from the cell cycle. These abnormalities contribute to early neuronal differentiation, cell migration, and a subsequent reduction in the number of neurons. Additionally, they demonstrated that knockdown of Kif23 results in abnormalities in the apical cell adhesion in the ventricular zone (VZ) of the E15.5 embryonic brain tissue. Finally, and most interestingly, they showed that the phenotype induced by Kif23 knockdown, where progenitor cells prematurely become neuronal cells and migrate to the Intermediate Zone, can be rescued by the expression of human Kif23. However, this rescue is not achieved with the mutated Kif23, which is known to cause microcephaly in humans.

Overall, the data suggest that Kif23 plays a crucial role in properly controlling the division orientation and cytokinesis of neural progenitor cells, thereby maintaining the proliferative capacity of progenitor cells. It is particularly interesting that the study provides insight into the mechanism by which mutations in Kif23, found in humans, can cause microcephaly. Most of the authors' conclusions are supported by sufficient quality and quantitative data, making the manuscript suitable for publication in EMBO J. However, some interpretations of the data are based on unclear evidence, and there are deficiencies in the presentation of statistical data. Therefore, I recommend that the following points be addressed or further explained before the manuscript is accepted for publication.

We thank the reviewer for providing the positive and constructive feedback, which helped us to significantly improve the quality of our manuscript.

Major points:

1. The authors claim that Kif23KD induces spindle disorganization. However, there is no specific description of the abnormalities and no clear criteria for determining "disorganized spindle." From the provided images (Fig4C, EV1), it is unclear what abnormalities are present. For instance, the spindle morphology in Fig4C appears slightly different from the control, but it might simply be due to the spindle being angled relative to the focal plane of the microscope. Given that the authors demonstrate changes in spindle angle due to Kif23KD in Fig4E-H, careful examination is needed to distinguish whether the observed spindle disorganization is due to viewing angle or actual morphological changes. Comparing spindles where gamma tubulin signals are visible at both poles could clarify this issue. Additionally, if significant spindle abnormalities occur in prometaphase/metaphase, the spindle assembly checkpoint should reduce the proportion of cells progressing to anaphase. If the authors strongly assert the occurrence of disorganized spindles, supporting data should be provided.

Response: We appreciate the reviewer's thoughtful feedback. Following the reviewer's suggestion, we have conducted further investigations to determine whether the abnormalities of spindle morphology could be attributed to *Kif23*-KD or simply to the angle at which it is viewed. We examined morphology of spindles in mitotic cells by triple staining of GFP, α -tubulin, and γ -tubulin. We selected the GFP⁺ metaphase cells with uniform γ -tubulin signals in both spindle poles and measured the intensity of α -tubulin. As the reviewer pointed out, there were no significant differences in α -tubulin intensity between the *Kif23*-KD and the control groups (**Fig. 5C, D**). This suggests that *Kif23*-KD does not affect the spindle organization during mitosis and may not trigger spindle assembly checkpoint activation. We have added new figures (**Fig. 5C, D**) and the corresponding description in the Results and Methods in the revised manuscript (p. 10, lines 232-236; p. 22, lines 594-596).

2. In Fig5I, the RhoA signal appears to be reduced regardless of GFP positivity. To substantiate the claim that Kif23KD leads to decreased RhoA expression, quantitative data is necessary. Furthermore, for the series of experiments shown in Fig5H, J, K, quantitative data is also required, as these experiments form a crucial part of the basis for the authors' conclusions.

Response: We appreciate the reviewer's constructive comment, and thus performed a quantitative analysis of apical RhoA mean fluorescence intensity, considering the intensity of RhoA within 30 μ m from the GFP⁺ ventricular surface as the apical RhoA intensity. We successfully observed a significant reduction in apical RhoA intensity in the *Kif23*-KD cortices compared to the control group (**Fig. 6D**). Additionally, we conducted a quantitative analysis of the mean fluorescence intensity of ECT2 in the spindle midzone in the GFP⁺ anaphase cells and the ratio of Cit-K labeled apical midbodies relative to the apical GFP⁺ cells. As expected, we noticed a significant reduction in ECT2 intensity in the spindle midzone (**Fig. 6B**) and Cit-K labeled apical midbodies (**Fig. 6G**) after *Kif23*-KD compared with the control group. We have included these new data in **Fig. 6B, D, G** and the Results in the revised manuscript (p. 10, lines 255-260).

3. The manuscript currently lacks information on the sample size for each experimental condition, and instead, the number of experimental repeats appears to be incorrectly denoted as "n=X". Typically, the number of repeats should be denoted as "N". Please include the sample size (n) for each group or condition. This information is essential for evaluating the statistical power and reproducibility of the results.

Response: We thank the reviewer for pointing out this critical issue. We now provide detailed information on the sample size and number of repeats in all the figure legends in the revised manuscript (p. 34-40).

Minor points:

- The figure citations on page 7 appear to be incorrect. Where it says (Figs. 2C, D), it should be (Figs. 2C, E), and where it says (Figs. 2E, F), it should be (Figs. 2D, F).
- On page 7, the text describing Fig2 states "at E15.5," but the figure and legend indicate "E16.5." Please correct this inconsistency.

Response: We thank the reviewer for noticing above discrepancy. We have carefully corrected the inconsistency in our revised manuscript (p. 7, lines 145, 151).

Dear Prof. Osumi,

Thank you again for the submission of your revised manuscript to The EMBO Journal. I apologize for the rather protracted review process on this occasion -which was due to the unavailability of the referees for a few weeks- and thank you very much for your understanding and patience.

We have now received the comments of both referees (included below for your information), and I am glad to say that they are both very satisfied with the revision, recognize that their previous concerns have all been completely and successfully addressed, and now recommend publication of the manuscript in The EMBO Journal. I am thus happy to say that your manuscript has been in principle accepted for publication in our journal. Congratulations on an excellent manuscript!

From the editorial side, there are a few minor changes and corrections that we need from you before we can proceed with formal acceptance of the manuscript and its publication:

- We noticed that the second (but not the first) co-author has been listed as co-corresponding (along with the last co-author), which is rather unusual. We kindly ask you to consider revising the order and/or correspondence author status taking into account the actual contributions of all co-authors. Please note that the detailed contributions of each co-author need to be specified in our manuscript handling system (please see relevant point below).
- Please note that the full funding information should be entered in our manuscript handling system (eJP) as well as listed in the Acknowledgements section of the manuscript itself. The funding information related to the "Support System for Young Researchers to use research equipment, instruments, and devices at Tohoku University Core Facility Center" is currently missing from eJP.
- Please remove the "Grant Sponsors" line from the second page of the manuscript.
- You may list up to 5 relevant keywords if you wish (you currently have 4). Please provide the list of keywords after the Abstract of your revised manuscript.
- Please move your Data Availability statement before the Acknowledgments section.
- Please note that at EMBO Press we encourage authors to include datasets obtained from public resources in the reference list in the form of data citations. More information on the format of such data citations and examples can be found in our guide: <https://www.embopress.org/page/journal/14602075/authorguide#referencesformat>.
- Please rename your Declaration of Interests statement to "Disclosure and competing interests statement".
- The author contributions statement should be removed from the manuscript file. Instead, we use CRediT to specify the contributions of each author in the journal submission system. Please feel free to use the free text box to provide more detailed descriptions during submission. See also our guide to authors for more information: <https://www.embopress.org/page/journal/14602075/authorguide#authorshippinguidelines>.
- Please add the heading "Appendix" at the top of the first page of your Appendix PDF file.
- Thank you for uploading your synopsis image. Please note that this image will be published online at the final dimensions: 550 pixels (width) x variable height (in the range 300-600 pixels). After resizing your image to a width of 550 pixels, we noticed that the text is hardly legible. Could you please increase the font sizes so that all text is easily readable at the final dimensions of the synopsis image?
- Please note that EMBO press papers are accompanied online by:
 - A) a short (2 sentences) summary (synopsis text) of the findings and their significance, and
 - B) 2-5 short bullet points highlighting the key results.Please upload this information in a separate Word file.
- Materials and methods need to be described in the manuscript using our Structured Methods format, which is now required for all research articles. According to this format, the Methods section includes a single "Reagents and Tools Table" -listing key reagents, experimental models, software and relevant equipment including their sources and relevant identifiers- followed by a "Methods and Protocols" section describing the methods. Please download and fill our Reagents and Tools Table template (.docx), which you can find in our author guide: <https://www.embopress.org/page/journal/14602075/authorguide#structuredmethods>. When submitting your revised manuscript, please do not include the Reagents and Tools Table in the Methods section of the manuscript but upload it as a separate file choosing the file type "Reagent Table".

- During our routine pre-acceptance checks, our data editors have raised the following queries regarding figures, data, and legends. You are kindly requested to address them all completely in your revised manuscript:

1. Please note that the legends for Figures 4b-c are not provided in the sequential manner (legend for Figure 4c is provided before legend of Figure 4b). This needs to be rectified.
2. Please note that the legend for Figure EV 3a is missing in the manuscript. This needs to be rectified.
3. Please note that the white dashed line in Figure 5g is incorrectly mentioned as black dashed line. This needs to be rectified.
4. Please note that the exact p values are not provided in the legends of Figures 2e-f; 3b-c, g; 4b, d, f, h, j; 5f; 6b, d, g; 7b, d, f; 8d; 9d; EV 1b-c.
5. Please note that information related to "n" is missing in the legends of Figures 1a-b; 9a.
6. Please note that the acronyms vz/svz/iz are not defined in the legend of Figure 3a; 6a, c, e-f. This needs to be rectified.

Please also note that as part of the EMBO publications' Transparent Editorial Process, The EMBO Journal publishes online a Peer Review File along with each accepted manuscript. This File will be published in conjunction with your paper and will include the referee reports, your point-by-point response and all pertinent correspondence relating to the manuscript. You can opt out of this by letting the editorial office know (contact@embojournal.org). If you do opt out, the Peer Review File link will point to the following statement: "No Peer Review File is available with this article, as the authors have chosen not to make the review process public in this case."

We look forward to seeing a final version of your manuscript as soon as possible. Please let us know if you have any questions and use this link to submit your revision: <https://emboj.msubmit.net/cgi-bin/main.plex>

Best wishes,

Ioannis

Referee #1:

The authors have addressed all my previous concerns at full satisfaction, so in my opinion this very nice paper is now ready for publication.

Referee #2:

The authors have addressed all my (Reviewer 2) comments by adding experimental data and providing clear explanations, leading to new insights.

All of my concerns have been thoroughly resolved, and they also seem to have taken an appropriate approach in responding to the comments of the other reviewer. The additional data have strengthened the support for the paper's conclusions, making the manuscript even more compelling.

I look forward to seeing this work published and shared with the broader community.

All editorial and formatting issues were resolved by the authors.

Dear Noriko,

Congratulations on an excellent manuscript! I am very pleased to inform you that it has been accepted for publication in The EMBO Journal. Thank you for your thorough responses to the initially raised referees' concerns, and for addressing all our editorial and formatting requests.

If you have any questions, please do not hesitate to contact the Editorial Office. Thank you for your contribution to The EMBO Journal. Working with you has been a pleasure!

Best regards,

Ioannis
